# Reconstructing landscapes : an adjoint model of the Stream Power and diffusion erosion equation

Carole Petit[1], Anthony Jourdon[1,2], and Nicolas Coltice[1]

[1]Université Côte d'Azur, CNRS, Observatoire de la Côte d'Azur, IRD, Géoazur, 250 rue Albert Einstein, Sophia Antipolis, 06560 Valbonne, France
[2]Sorbonne Université, CNRS, Institut des Sciences de la Terre de Paris, ISTeP, F-75005 Paris, France

**Correspondence:** Carole Petit (carole.petit@univ-cotedazur.fr)

**Abstract.** We simulate landscape evolution using a diffusion-advection equation with a source term, where the advection velocity is derived from the classical parametrization of the Stream Power Law. This formulation allows for forward modeling of uplift, hillslope and fluvial erosion within a finite-element framework, and enables the use of adjoint methods for sensitivity analysis and parameter inversion. When considered individually, model parameters such as the diffusion coefficient, fluvial erodibility, initial topography, and time-dependent uplift can be inverted using constraints from final topography, sediment flux, or cumulative denudation at specific locations. Sensitivity analysis on a real landscape reveals that sensitivity to erosion parameters is higher in steep, high-relief areas and that hillslope diffusion and fluvial incision affect the model differently. After a series of tests on synthetic topographies, we apply the adjoint model to two natural cases: (1) reconstructing the pre-incision topography of the southeastern French Massif Central, which appears as a smooth, flat footwall bounded by a linear escarpment along a major lithological boundary; and (2) estimating the Quaternary uplift rate along the Wasatch Range, USA, where our model suggests a significant increase in uplift from 0.2 to 1 $\mathrm{mm.yr^{-1}}$ over the last $\sim$2 million years.

## 1 Introduction

Tectonic events coupled to climatic variations shape the Earth's surface over timescales ranging from thousands to millions of years. While modern climate, topography, and fluvial activity can be recorded and monitored directly, reconstructing past landscape evolution remains challenging. To establish relationships between lithological variations, uplift, and erosion-sedimentation mechanisms that quantify sensitivity to fluvial incision or hillslope processes, Landscape Evolution Models (LEMs) have been developed (e.g., Tucker and Slingerland, 1994; Tucker et al., 2001; Davy and Lague, 2009; Tucker and Hancock, 2010; Braun and Willett, 2013; Carretier et al., 2016; Salles and Hardiman, 2016).

Most LEMs are forward models: they rely on assumptions about initial and boundary conditions and are retrospectively validated against available data such as surface morphometry, sedimentation, exposure age dating, or low-temperature thermochronology. These models typically include numerous parameters to capture the influence of climate, lithology, sediment granulometry, isostasy, and/or tectonics on landscape evolution (Carretier et al., 2016; Salles, 2016; Braun and Willett, 2013; Tucker et al., 2001). As the number of parameters increases to account for various forcings and physical processes, evaluating the influence of each parameter on the model outcome becomes increasingly complex. Moreover, geomorphological obser-

vations rarely provide clear constraints on model parameters, and non-uniqueness arises because different parameter sets can produce similar "final" topographies. Model formulation and implementation also plays a role: the choice of erosion, transport, sedimentation, or tectonic laws can significantly affect results and complicate sensitivity analysis and model validation (Barnhart et al., 2020b).

Inverse modeling offers a way to address these issues by estimating model parameters or histories directly from observations. Although still less common in geomorphology than in other geosciences disciplines, inverse approaches have been widely developed over the past decades. Neighbourhood algorithms have proven useful to constrain erosion parameters, uplift history, or to reconstruct ancient (pre-glacial) landscapes by fitting the present-day topography and/or sediment fluxes (e.g., Croissant and Braun, 2014; Pedersen et al., 2018). More complex modelling frameworks combine erosion with cosmogenic isotope production or low-temperature thermochronology to further constrain scenarios of landscape evolution (e.g., Glotzbach, 2015; Braun, 2003; Braun et al., 2012). Bayesian approaches have also been employed to infer erosional histories from sedimentary records (Yuan et al., 2019), or to explore temporal variations in parameters and their uncertainties (Chandra et al., 2019). Despite their methodological differences, these approaches rely on extensive sampling of the parameter space and thus require very large numbers of forward simulations.

Among inverse models, many studies have been devoted to inferring uplift rates or climatic histories from river longitudinal profiles (Goren, 2016; Roberts et al., 2012; Roberts and White, 2010; Petit et al., 2017) because of their particular interest: they progressively adjust to external forcings until reaching a steady state (Willett and Brandon, 2002). The response time, defined as the period required for the channel profile to adjust to new conditions (Armitage et al., 2013; Godard et al., 2013), controls how long river channels can preserve a record of past conditions. This framework has been applied to explain large-scale landscape features and major channel knickzones, such as those observed in Colorado, Australia, and Africa, as the result of long-term uplift rate variations (Roberts et al., 2012; Roberts and White, 2010; Rudge et al., 2015). Provided the uplift rate is the only unknown and spatially uniform, the channel slope is directly related to the uplift rate (Royden and Perron, 2013), making the inversion relatively straightforward. All these studies highlight the potential of inverse modeling to advance our understanding of landscape evolution, improve parameter estimation, and quantify uncertainties. However, beyond river profiles, most inverse approaches remain constrained by computationally intensive parameter-space exploration. The same challenge affects sensitivity analysis, which seeks to quantify how model parameters and formulations influence outcomes. For example, the Morris method has been used to test sensitivity of the CAESAR-Lisflood model over short timescales ($\sim$10 years) by perturbing parameters individually and simplifying outputs into response functions (Skinner et al., 2018). Extensions of this approach to multi-model frameworks and longer timescales (13 ka) show how including various processes and increasing model complexity shape outcomes and performance (Barnhart et al., 2020b, c). Similarly, Armitage et al. (2018) demonstrated that long-term (Myr-scale) sensitivity to precipitation changes depends strongly on whether sediment transport is described by diffusion or incision laws influencing the model response time to perturbations and the outgoing sediment flux. Such studies provide valuable insights but often at a high computational cost.

In contrast to the previously cited gradient-free methods, the adjoint approach enables efficient sensitivity analysis and the inversion of unknown parameters by a cost-effective, gradient-based approach (e.g., Hu and Kozlowski, 2020). Despite their

potential and while already used in other fields such as turbidity currents modeling (Parkinson et al., 2017), terrain correction for weather forecasting (Tao et al., 2019) or hydrodynamics (Clare et al., 2022) adjoint approaches remain underexplored in geomorphology. The adjoint method involves solving an additional partial differential equation (PDE), called the adjoint equation, that quantifies how variations in the model's output variables affect the objective function (or cost function) that is being optimized (e.g., Givoli, 2021). By solving both the forward and adjoint problems, the gradient of the cost function can be computed efficiently with respect to all parameters simultaneously, regardless of their number. The resulting sensitivity is local, describing the gradient of the cost functional in the vicinity of the current parameter state.

In this paper, we introduce a new modelling approach that simulates fluvial incision and hillslope processes as a diffusion-advection equation discretized with the finite-element (FE) method. The adjoint equation is solved to estimate model sensitivity to erosion parameters and to reconstruct the initial conditions, source term or erosion coefficients in different settings. Our implementation combines components from two open-source tools: (1) Landlab (Hobley et al., 2017; Barnhart et al., 2020a; Hutton et al., 2020), which we use to to remove local topographic minima and compute the drainage network and area, and (2) Firedrake (Ham et al., 2023), a Python-based software framework for automating the solution of partial differential equations using the finite element method, which we use for FE discretization, and numerical solution of the forward and adjoint problems.

We demonstrate the approach on two natural examples with relatively simple geological settings and available data such as denudation, exhumation, and uplift rates. The first case study is the southeastern border of the Massif Central, where Late Miocene to Pliocene uplift triggered intense fluvial incision along the Cévennes Fault escarpment (Olivetti et al., 2016; Fauquette et al., 2020; Séranne et al., 2021). Reconstructing the pre-incision landscape could provide insights into escarpment geometry and initial conditions for further forward models investigating retreat mechanisms. The second case is the Wasatch Fault in the northeastern Basin and Range, a well-studied active normal fault with uplift rates ranging from 0.3 to 1.7 $\mathrm{mm.yr}^{-1}$, depending on timescale, location and methodology (Ehlers et al., 2003; Friedrich et al., 2003; Mayo et al., 2009; Stock et al., 2009; Smith et al., 2024). Despite extensive data, uncertainties remain about the long-term evolution of footwall uplift since the Miocene (Armstrong et al., 2003; Ehlers et al., 2003). Together, these examples allow us to test the potential and limitations of adjoint-based landscape inversion in natural settings.

## 2 Methods

### 2.1 Simulating landscape evolution with a diffusion-advection equation

Landscape evolution models aim at solving a conservation equation (e.g., Simpson and Schlunegger, 2003; Perron et al., 2008):

$$\frac{\partial h}{\partial t} = -[\nabla \cdot (\mathbf{Q_h}) + \nabla \cdot (\mathbf{Q_f})] + U, \tag{1}$$

where $h$ is the elevation [L], $\partial h / \partial t$ is the rate of change of elevation over time [L.T$^{-1}$], $\mathbf{Q_h}$ and $\mathbf{Q_f}$ are sediment fluxes per unit surface [L$^2$.T$^{-1}$] coming from hillslope processes and fluvial incision, respectively, and $U$ is the uplift rate [L.T$^{-1}$]

accounting for tectonic and/or isostatic vertical movements. The flux $\mathbf{Q_h}$ represents hillslope processes, which are responsible for the progressive smoothing of the landscape with time and is modelled as a diffusion process (e.g., Armstrong, 1987):

$$\mathbf{Q_h} = -\kappa\nabla h, \tag{2}$$

with $\kappa$ the diffusion coefficient [$\mathrm{L}^2.\mathrm{T}^{-1}$]. Physical and empirical relationships have permitted to relate the fluvial incision rate arising from the divergence of the fluvial sediment flux $\mathbf{Q_f}$ to the local channel slope $S$ and the upstream drainage area $A$ [$\mathrm{L}^2$]. This relationship, known as the Stream Power Law (SPL) (e.g., Seidl et al., 1994; Whipple and Tucker, 1999), simulates the incision of bedrock river channels as:

$$\nabla \cdot \mathbf{Q_f} = K_f A^m S^n, \tag{3}$$

with $K_f$ the rock erodibility [$\mathrm{L}^{1-2\mathrm{m}}.\mathrm{T}^{-1}$], $m$ an exponent related to the area, and $n$ an exponent controlling the non-linearity of the fluvial erosion (e.g., Howard et al., 1994). It assumes that river incision is detachment-limited, meaning the erosive power of the fluvial system is controlled solely by the flow's capacity to detach material from the bedrock. The local channel slope $S$ is equal to the norm of the topographic gradient vector $\nabla h$ such that Eq. (3) can be written:

$$\nabla \cdot \mathbf{Q_f} = K_f A^m \|\nabla h\|^n. \tag{4}$$

Substituting Eqs. (2) and (4) into Eq. (1) yields:

$$\frac{\partial h}{\partial t} = \nabla \cdot (\kappa\nabla h) - K_f A^m \|\nabla h\|^n + U. \tag{5}$$

Eq. (5) can be cast into a diffusion-advection equation (e.g., Perron et al., 2008; Braun and Willett, 2013) considering a few modifications:

1. The exponent $n$ is set to 1 to linearize the dependence of the advection term to the topographic gradient (e.g., Seidl et al., 1994). Morphometric analyses provide estimates of the concavity index $\theta = m/n$ (Hack, 1957) (usually around 0.5) which is used to (1) linearize the advection term ($n = 1$) and (2) estimate the parameter $m$.

2. The scalar quantity $K_f A^m$ is interpreted as the norm of a velocity vector

$$\mathbf{c} := K_f A^m \mathbf{u}, \tag{6}$$

where $\mathbf{u}$ is a unit vector oriented along the topographic gradient:

$$\mathbf{u} := \frac{\nabla h}{\|\nabla h\|}. \tag{7}$$

This velocity, $\mathbf{c}$, represents the rate of propagation of the topographic information e.g., the rate of knickpoint propagation within the drainage network (Whipple and Tucker, 1999).

Thus, using the definition of Eq. (6) and setting $n = 1$, Eq. (5) becomes:

$$\frac{\partial h}{\partial t} - \nabla \cdot (\kappa \nabla h) + \mathbf{c} \cdot \nabla h = U. \tag{8}$$

We note that Eq. (8) has the form of divergence-free advection-diffusion equation, which directly arises from the use of the SPL (see Appendix A).

Assuming that the local channel slope is equal to the norm of the topographic gradient requires that the direction of the gradient is accurately respected when computing the river network. Significant approximations in the gradient direction would limit the validity of Eq. (5) when interpreted as a diffusion-advection equation. In many LEMs however, due to the discretization of the topographic grid and the use of single-flow direction water routing algorithms, the computed drainage direction is not always co-linear with the topographic gradient as it should be. Certain routing algorithms, such as the $D_\infty$ method (Tarboton, 1997), distribute water flux between two nodes with links oriented closely to the gradient direction, which permits to minimize the orientation discrepancy between the drainage direction and the true topographic gradient. The $D_\infty$ method also permits divergent drainage to occur, which is impossible with single-flow direction methods.

## 2.2 Solving the forward problem

Solving equation Eq. (8) as a classical diffusion-advection PDE offers several advantages. First, in a diffusion-advection equation, the quantity which is involved in diffusive and advective processes is expected to be the same i.e., the gradient $\nabla h$ (Simpson and Schlunegger, 2003). In contrast, in Eq. (5) the diffusive term uses the actual topographic gradient, while the fluvial incision term uses the local channel slope, which, depending on the local problem and mesh resolution, may or may not correspond to the topographic gradient. Moreover, this new approach naturally propagates information upstream by advecting topographic characteristics with a velocity proportional to the drainage area in the direction of the gradient resulting in two important features: (1) along channel sides, altough slow, the advection velocity can be nonzero, allowing channel widening with time and (2) along channel paths, despite small slopes, the advection is more pronounced thanks to the large drainage area.

Furthermore, the classical coupling of 2D hillslope diffusion with 1D (vertical) fluvial incision in Eq.(5) may introduce scaling issues and strong resolution dependency, arising from the implicitly fixed river width (Hergarten, 2020). Reformulating the model as Eq.(8) helps reduce these issues: while resolution still affects drainage area computation and topographic gradient estimation, the assumption of a fixed channel width is no longer required, since vertical incision is reinterpreted as horizontal advection. Finally, this formulation provides a solid basis for deriving and solving the adjoint equation associated with Eq. (8).

In this study, we present a new method to both solve the forward (Eq. 8) and adjoint PDEs describing landscape evolution as a classical diffusion-advection equation, with a time- and space-variable advection velocity. At each time step $k$, the topography from the previous step $k - 1$ is used to compute the drainage area using Landlab's $D_\infty$ flow routing algorithm. The topographic gradient $\nabla h$ is calculated using the finite elements derivative operators, normalized to obtain the unit vector $\mathbf{u}$, and the advection velocity vector $\mathbf{c}$ is finally computed according to Eq. (6). This updated advection velocity is then incorporated into the weak formulation of Eq. (8), which is then solved using Firedrake's finite element framework with $Q_1$ elements.

To solve Eq. (8), the system is closed with the initial and boundary conditions:

$$h(\mathbf{x}, t_0) = h_0 \qquad\qquad \forall \mathbf{x} \in \Omega \text{ and } t_0 = 0,$$

$$h(\mathbf{x}, t) = \bar{h} \qquad\qquad \forall \mathbf{x} \in \partial\Omega \text{ and } \forall \mathbf{t} \in [0, T],$$

where $\Omega$ is the model domain, $\partial\Omega$ its boundary, $h_0$ the initial topography, and $\bar{h}$ the boundary conditions. The latter are of Dirichlet type (fixed elevation), and can be updated over time to account for progressive uplift at the domain boundaries.

During the forward model run, the time-dependent sediment flux is recorded as the cumulative volume of eroded material, which can subsequently be used in the adjoint model as an additional constraint for parameter inversion. Local denudation, defined as the cumulative height of eroded material at each node over time, can also be recorded.

## 2.3 Adjoint method

### 2.3.1 Governing equations

We consider the PDE describing the evolution of surface elevation $h$ as in Eq. (8). We rewrite this equation in the residual form $Lh - f = 0$, where: $L := L(p)$ is a differential operator acting on $h$, such as:

$$Lh = \frac{\partial h}{\partial t} + \mathbf{c} \cdot \nabla h - \nabla \cdot (\kappa \nabla h), \qquad\qquad (9)$$

and $f$ is the source term, which in this case corresponds to the uplift rate $U$. The solution $h := h(p)$ of this PDE is controlled by a set of parameters $p$ (initial conditions, $h_0$, velocity, $\mathbf{c}$, diffusivity, $\kappa$, and source term, $U$).

We define a cost functional $J := J(h)$ that has to be minimized:

$$J = \int_{\Omega} \phi(h) \mathrm{d}V, \qquad\qquad (10)$$

where $\phi(h)$ can be defined as the residual between output topography at time $T$ and observations $h_{obs}$ such that:

$$\phi(h_T) = \frac{1}{2}(h_T - h_{obs})^2, \qquad\qquad (11)$$

and its derivative with respect to $h$ is:

$$\phi'(h) = \frac{\mathrm{d}\phi}{\mathrm{d}h} = h - h_{obs}. \qquad\qquad (12)$$

The objective is to evaluate the gradient $\frac{dJ}{dp}$, which will then allows us either to minimize $J$ (for parameter optimization) or to analyze the influence of certain parameters on the model result (for sensitivity analysis). To this end, we first solve the forward problem using trial parameter values $p$, and then compute the adjoint variable $\lambda$ by solving the adjoint equation:

$$L^{\dagger}\lambda = \phi', \qquad\qquad \forall \lambda \in \Omega, \qquad\qquad (13)$$

$$\lambda = 0 \qquad\qquad \forall \lambda \in \partial\Omega. \qquad\qquad (14)$$

Here $L^\dagger$ is the formal adjoint of the operator $L$. For diffusion-advection equations, the adjoint operator has the same form as the forward operator, except that the advection velocity $\mathbf{c}$ is reversed (e.g., Celia et al., 1990):

$$L^\dagger \lambda = \frac{\partial \lambda}{\partial t} - \mathbf{c} \cdot \nabla \lambda - \nabla \cdot (\kappa \nabla \lambda). \tag{15}$$

First, we have to solve for $\lambda$ in Equation 13, where the forcing term is $\phi'$. Once the adjoint variable $\lambda$ is computed, we use it to evaluate the gradient of the cost functional with respect to the parameters $p$:

$$\frac{\mathrm{d}J}{\mathrm{d}p} = -\int_\Omega h \frac{\partial L^\dagger}{\partial p} \lambda \mathrm{d}V. \tag{16}$$

The partial derivative $\frac{\partial L}{\partial p}$ can be computed analytically when possible, or numerically. The functional $J$ may also include time-dependent controls, in which case it will integrate the misfit between models and observations over the model duration $T$.

The computational workflow is as follows:

1. Solve the forward problem $Lh - f = 0$ for a given set of parameters $p$,

2. Solve the adjoint problem for $\lambda$ using Equation (13),

3. Compute the gradient $\frac{\mathrm{d}J}{\mathrm{d}p}$ using Equation (16),

4. Update $p$ using an optimization algorithm, and iterate until convergence. Convergence is assumed when the residual
value stabilizes, i.e., its relative change between successive iterations is less than $0.1\%$, or when the maximum number of iterations (typically 50) is reached.

In practice, the Firedrake package automates this process using automatic differentiation (AD) techniques. The user only needs to define the weak form of the PDE and specify the functional $J$.

Beyond parameter inversion, the adjoint directly provides the model relative (dimensionless) sensitivity to a parameter $p$
through:

$$S_p^{\mathrm{rel}} = \frac{p}{J} \frac{\partial J}{\partial p}, \tag{17}$$

where all physical quantities are non-dimensionalized to ensure consistent scaling of both residuals and sensitivities. This relative sensitivity metric is dimensionless by construction, enabling direct comparison across parameters of different nature and scale, and supporting robust sensitivity analysis.

### 2.3.2   Definition of the cost functional

In our problem, the objective function $J$ combines data misfit terms (comparing modeled and observed topography, sediment flux, or denudation rates) with regularization components. To recover solutions with sharp discontinuities while preserving smooth regions, we employ total variation (TV) regularization (Strong and Chan, 2003). Specifically, for the inversion of

initial conditions, we replace the standard $L_2$-norm used in Eqs. (10) and (11) with an edge-preserving TV regularization scheme, adapted from image processing methods (Barcelos, 2002).

$$J = \int_\Omega \left[ \alpha g \|\nabla h\| + \frac{1}{2}(1-g)(h - h_{\text{obs}})^2 \right] dV, \tag{18}$$

$$g = \left(1 + k|\nabla G * h|^2\right)^{-1}, \tag{19}$$

where $\alpha > 0$ controls the regularization strength, $g \in [0,1]$ is an edge-detection function using Gaussian-filtered ($G$) topography gradients, and $k$ modulates edge sensitivity. We calibrate the parameters $\alpha$ and $k$ to suppress high-frequency noise in reconstructed solutions and preserve geomorphologically significant sharp features (e.g., fault scarps, knickpoints). Increasing $\alpha$ smoothes the residual and reduces short-wavelength peaks, while larger values of $k$ tend to merge together areas of similar values. Finally, we define the functional $J_{tot}$ which incorporates regularization terms and misfits to sediment flux and denudation values as follows:

$$J_{tot} = J + C \sum_{i=0}^{t_{max}} (q_i - q_{i_{obs}})^2 \Delta t + D \sum_{j=0}^{N} \sum_{i=0}^{t_{max}} (d_{ij} - d_{ij_{obs}})^2 \Delta t + \text{regularization terms} \tag{20}$$

The first term of the functional represents the misfit between modelled and observed topography integrated over the entire domain. The second term incorporates a penalization of the differences between the estimated outgoing sediment flux, $q$, and the true sediment flux, $q_{obs}$, summed over discrete time intervals corresponding to the model timestep $\Delta t$. The third term represents the difference between observed and modelled cumulated denudation, $d$, with time on some prescribed nodes $n$ that could correspond to sampling points. Finally, the coefficients $C$ and $D$ control the relative importance of the sediment flux and denudation residual terms in the inversion. Depending on the parameter we want to analyse, coefficients $C$ and/or $D$ may be zero.

## 2.4 Application to synthetic and natural landscapes

We first conduct synthetic experiments by running first the forward FE model to generate synthetic data (e.g., topography, sediment flux, and/or cumulative denudation over time). We then use these data as controls for sensitivity analysis and/or inversion of the main model parameters: initial conditions, erodibility, diffusion coefficient, and source term. In the adjoint method, the forward model is executed once, starting with an initial guess that is slightly different from the value used for generating the synthetic data. In certain inversion procedures, we re-run the forward model using the inverted parameters to compare model outputs (topography, sediment flux, etc.) with synthetic data. Initial topographic conditions for the forward model can consist of simple geometric shapes (linear drainage divide, linear escarpment) or can be derived from an existing digital elevation model (DEM). For instance, to evaluate the model's ability to reconstruct spatial variations of $\kappa$ and $K_f$, we use as initial conditions the same DEM as in section 3.1.4, i.e., the south-east border of the French Massif Central. This region provides a suitable test case for highlighting spatial model sensitivity to erosion processes, as it displays a contrasted topography with both flat, smooth surfaces and deeply incised slopes. In both cases, we extract the target topography from the Shuttle Radar Topography Mission database (NASA, 2013) and re-sample it at the desired model resolution.

**3  Results**

| Model n° | IC | Dimensions, Resolution m², elem | Type | Duration | $\kappa$ m$^2$.yr$^{-1}$ | $K_f$ yr$^{-1}$ | $U(t)$ mm.yr$^{-1}$ | Figure n° |
|---|---|---|---|---|---|---|---|---|
| 1 | Square | $(19 \times 10^4)^2$, $128 \times 128$ | Analysis | 1 Myrs | $0.5(\mathbf{1}) \times 10^{-2}$ | $5 \times 10^{-7}$ | 0 | 1b |
| 2 | Square | $(19 \times 10^4)^2$, $128 \times 128$ | Analysis | 1 Myrs | $10^{-2}$ | $1(\mathbf{0.5}) \times 10^{-6}$ | 0 | 1c |
| 3 | Square | $(19 \times 10^4)^2$, $128 \times 128$ | Inversion | 1 Myrs | X | $5 \times 10^{-7}$ | 0 | 2 |
| 4 | Square | $(19 \times 10^4)^2$, $128 \times 128$ | Inversion | 1 Myrs | $10^{-3}$ | X | 0 | 3 |
| 5 | X (**Scarp**) | $(4 \times 10^4)^2$, $128 \times 128$ | Inversion | 1 Myrs | $5 \times 10^{-3}$ | $3 \times 10^{-6}$ | 0 | 4 |
| 6 | Square | $(4 \times 10^4)^2$, $128 \times 128$ | Inversion | 4 Myrs | $5 \times 10^{-3}$ | $4 \times 10^{-6}$ | X (**0.5**) | 5 |
| 7 | X | $(1.9 \times 10^5)^2$, $128 \times 128$ | Inversion | 4 Myrs | $10^{-3}$ | $10^{-6}$ | 0 | 6 |
| 8 | Divide | $2.2 \times 10^4 \times 1.4 \times 10^4$, $128 \times 80$ | Inversion | 4 Myrs | $10^{-1}$ | $3 \times 10^{-6}$ | X | 7 |
| 9 | Square | $(4 \times 10^4)^2$, $128 \times 128$ | Taylor test | 0.5 Myrs | $10^{-2}$ | $3 \times 10^{-6}$ | 0.5 | B1 |

**Table 1.** Summary of models and parameters. IC denotes initial conditions; resolution is the number of elements in the x and y dimensions. For inverse models, the letter X indicates which parameters has been inverted. For models 1 and 2, the parameter value indicate the guess value for which the sensitivity has been computed. Numbers in bold within parentheses indicate which values were used in the corresponding forward model (if relevant).

## 3.1 Sensitivity analysis and inversion

### 3.1.1 Erosion parameters

We first evaluate the model sensitivity and its capability to reconstruct spatially variable erosion coefficients $K_f$ and $\kappa$ on a $190\,\mathrm{km} \times 190\,\mathrm{km}$ grid with $128 \times 128$ elements. The initial topography is similar to the target topography used in Section 3.1.4 for initial condition inversion — specifically, a dissected NE-SW linear escarpment marking the southeastern boundary of the French Massif Central (Fig. 1a, 2, and 3).

In the first test, we assume spatially uniform values of $K_f = 5 \times 10^{-7}$ $\mathrm{yr}^{-1}$ and $\kappa = 10^{-2}$ $\mathrm{m^2.yr^{-1}}$, and independently compute the relative model sensitivity $S_p^{\mathrm{rel}}$ to each parameter, starting with trial values of $1 \times 10^{-6}$ $\mathrm{yr}^{-1}$ and $5 \times 10^{-3}$ $\mathrm{m^2.yr^{-1}}$, respectively. The $m$ exponent on the drainage area in Eq. 6 is always equal to 0.5. The sensitivity maps in Figs. 1b and 1c represent the spatial variation of the gradient of the cost functional with respect to these 2 parameters at the point defined by the trial value in the parameter space. The sensitivity values for $\kappa$ are generally low, except in two distinct regions: the NW corner of the domain, where the Cenozoic Cantal stratovolcano is located, and a NE-SW elongated zone corresponding to the southeastern escarpment of the Massif Central (Fig. 1b). The model sensitivity to the $K_f$ coefficient is larger, especially in the Massif Central region, and is even most pronounced around the headwaters of the major catchments, where topographic slopes are steep (Fig. 1c). Overall, both tests suggest that the model's sensitivity to erosion coefficients is particularly large along the NE-SW trending southern Massif Central escarpment, and very small in the low altitude areas of the Rhône plain.

We then prescribe a spatial variation of one coefficient (while keeping the other spatially uniform) on a checkerboard grid with $100 \times 100$ km squares and run the forward models for 1 Myrs. The source term is null, meaning that no uplift is imposed to the surface topography. In the inversion tests, we fix the model duration, initial conditions, source term and either the erodibility or the diffusion coefficient as identical to that of the forward models. Using these known values as constraints, we then perform separate inversions to estimate the unknown parameter. The main goal is to evaluate the model's ability to accurately recover the imposed spatial variations of $K_f$ or $\kappa$ (see Table 1).

We carry out the inversion of the diffusion coefficient $\kappa$ using only the final topography as the reference "observable" ($h_{obs}$) to control the inversion. Since hillslope diffusion is a local process, it would not be interesting to use other data representing long-distance sediment transport, such as the outgoing sediment flux, as a control. The "true" $\kappa$ field used in the forward model is defined as a checkerboard pattern with alternating low values of $10^{-2}\mathrm{m^2.yr^{-1}}$ and high values of $5 \times 10^{-2}\mathrm{m^2.yr^{-1}}$ (Fig.2a). At the start of the inversion, $\kappa$ is initialized everywhere to the mean of these two values. The inversion of the diffusion coefficient $\kappa$ converges after 33 iterations and the resulting $\kappa$ map (Fig. 2b and 2c) shows that the inversion correctly retrieves the spatial variations of the $\kappa$ coefficient, with some lateral smearing. In regions of very low sensitivity, such as the southeastern corner of the grid, the model fails to recover the checkerboard pattern.

The inversion of the erodibility coefficient $K_f$ is performed using both the final topography and the time-distributed outgoing sediment flux as observables. In the forward model, $K_f$ is prescribed as a checkerboard pattern with alternating low values of $10^{-6}\mathrm{yr}^{-1}$ and high values of $2 \times 10^{-6}\mathrm{yr}^{-1}$ (Fig. 3a). As in the $\kappa$ inversion, the initial guess for $K_f$ is set to the mean of these two values. The inversion converges after 31 iterations, producing a $K_f$ map that successfully recovers the checkerboard

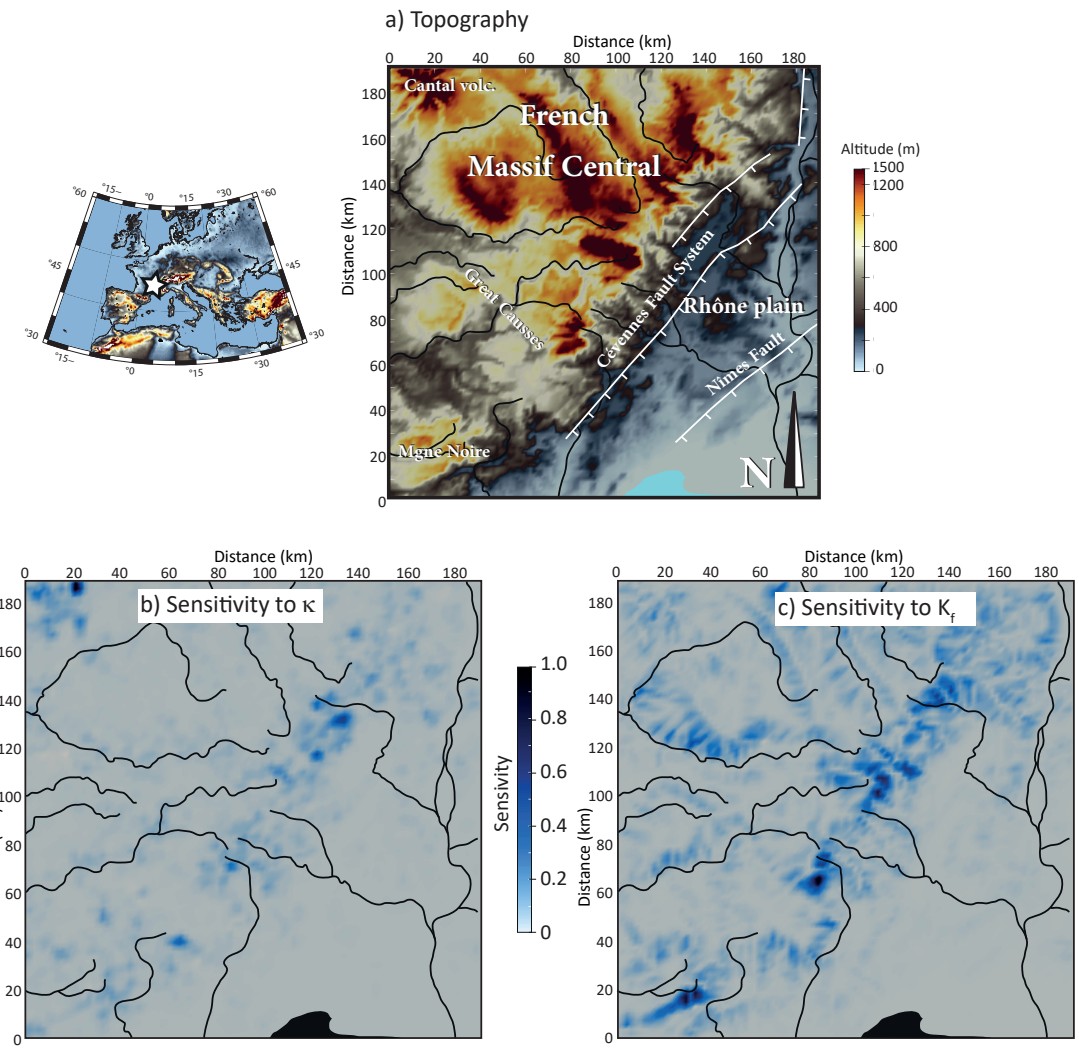

**Figure 1.** Sensitivity tests on a real topography for the diffusion coefficient $\kappa$ and the erodibility coefficient $K_f$ (models 1 and 2, Table 1). (a): Topography of the SE part of the French Massif Central with the main drainage systems (solid lines). White star on the left map indicates the location of the study area. (b): Relative model sensitivity (absolute value) to $\kappa$. (c): Relative model sensitivity (absolute value) to $K_f$.

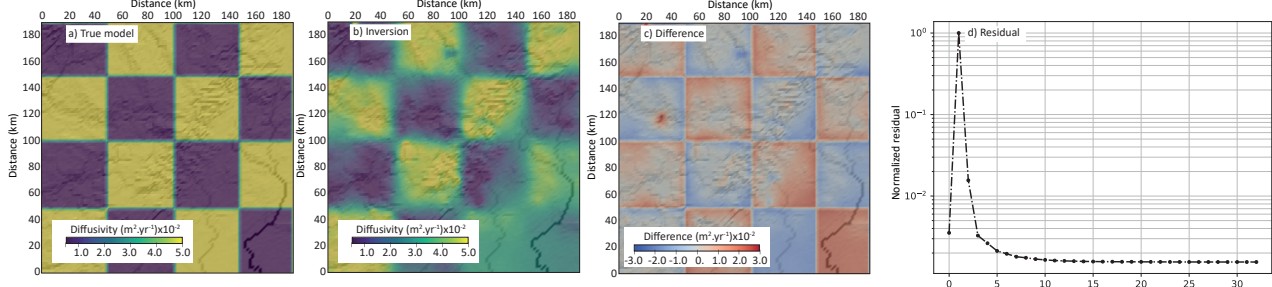

**Figure 2.** Inversion of the diffusion coefficient $\kappa$ with the adjoint method (model 3, Table 1). (a): True spatial variations of $\kappa$ used to build the forward model. (b): inverted $\kappa$ values map. (c): Residual. (d): Inverse model convergence. Same initial topography as on Figure 1.

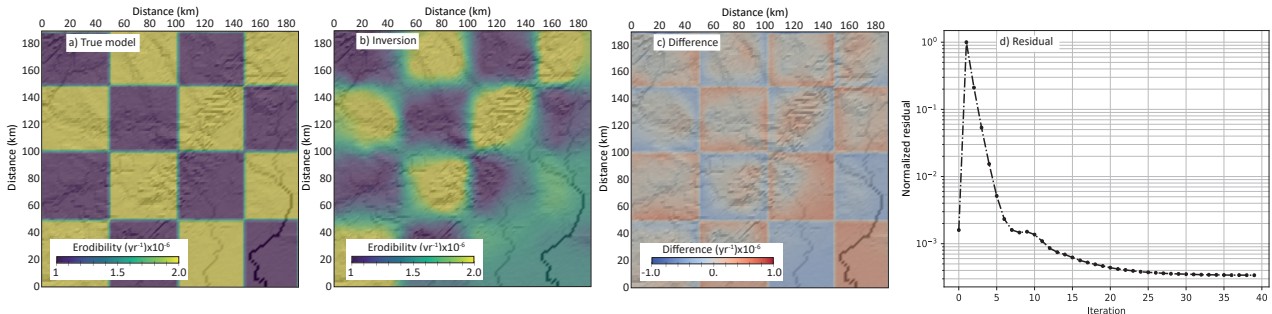

**Figure 3.** Performance of the inversion of the erodibility coefficient $K_f$ with the adjoint method (model 4, Table 1). (a): True spatial variations of $K_f$ used to build the forward model. (b): inverted $K_f$ values map. (c): Residual. (d): Inverse model convergence. Same initial topography as on Figure 1.

pattern (Figs. 3b and 3c). However, as with the $\kappa$ case, the southeastern corner of the model domain, where sensitivity is low, shows poor reconstruction of the imposed pattern.

### 3.1.2 Initial conditions

In this section, we demonstrate the model ability to recover initial conditions when erosion parameters ($K_f$, $\kappa$, $m$), uplift rate and erosion duration are known. We generate a synthetic topography with a simulation lasting for 1 million years on an initial topography featuring a linear escarpment associated with a random topographic noise of maximum 40 meters, and no uplift (Fig. 4a). The final topography resulting from this simulation shows a strongly incised escarpment, though the surface trace of the fault is still relatively easy to follow. It is then used as the reference observable for the adjoint model, together with the evolution of the outgoing sediment flux with time.

The inversion process starts with an initial guess for the initial conditions, which consists of a linear topographic slope in the same direction as in the forward model, but devoid of any escarpment (Fig. 4b). Starting from this initial guess, the

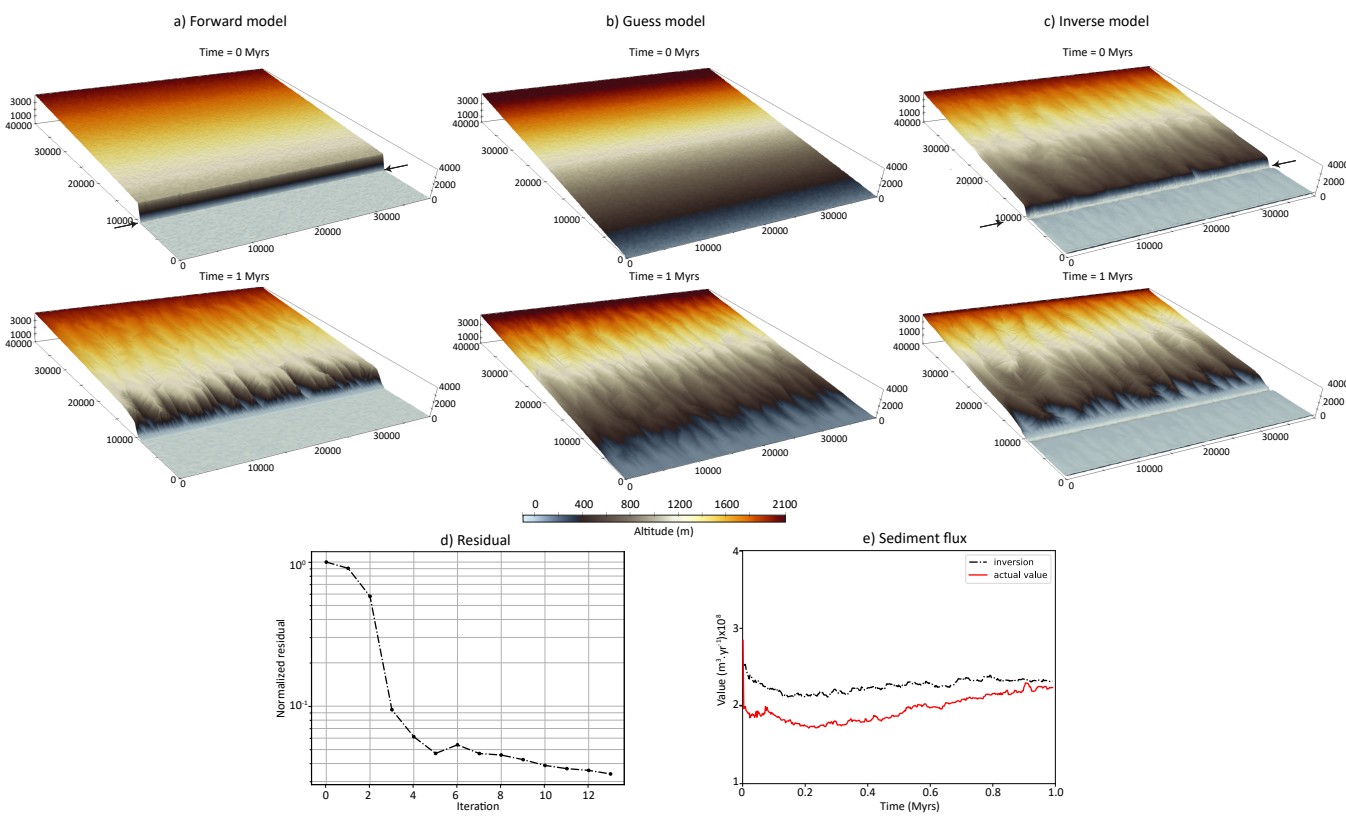

**Figure 4.** Inversion of initial conditions on a south-facing escaprment (model 5, Table 1). Top and middle panels: initial and final (1 Myrs) topographies in the forward model, the initial guess and the inversion result (a, b and c, respectively). Bottom panels: Relative residual (d) and modelled sediment flux (e).

inversion successfully reconstructs an initial topography featuring a well-defined escarpment (Fig. 4c). This inverted initial topography deviates from the initial guess and exhibits two smooth surfaces separated by a steep escarpment located close to the actual one, on which the notches produced during the 1 Myr incision history have been completely removed. Some undulations remain visible on the upper plateau, and the amplitude of the inverted escarpment is slightly lower than that of the true topography. The result is obtained after 13 iterations with a coefficient $\alpha = 5 \times 10^{-3}$ and an edge preservation coefficient $k = 200$ in Eq. (18). The normalized residual stabilizes around $3 \times 10^{-2}$ and the modelled sediment flux is consistent the actual one although slightly higher (Fig. 4d and e). This discrepancy likely arises because the inverted initial topography is not perfectly flat upstream, which promotes the rapid development of a drainage network and consequently enhances erosion and increases the outgoing sediment flux.

### 3.1.3 Source term

We now aim to assess the adjoint method's ability to accurately estimate source term variations, assuming that the boundaries of the uplifting area are known. The forward model simulates a square mountain, similar to Model 1. In this scenario, the mountain experiences a constant uplift rate of $0.5 \ \mathrm{mm.yr^{-1}}$ over 4 million years.

In the inversion process, we use the final topography, outgoing sediment flux and cumulative denudation on 3 specific model points as control points on the inversion (Fig. 5a). We run the forward model for a duration of 4 Myrs using a time step of 2000 years. The topography modelled with the inverted uplift rate value after 4 Myrs is very similar to the forward model 1, although slightly higher (Fig. 5a and b). The inversion converges after a few iterations and the results indicate that the uplift rate value is correctly estimated, with a value ranging from $0.53 \ \mathrm{mm.yr^{-1}}$ at the beginning to $0.51 \ \mathrm{mm.yr^{-1}}$ at the end of the model (i.e., with a maximum error of 6 %), with a significant deviation from the initial guess value of $0.2 \ \mathrm{mm.yr^{-1}}$ (Fig. 5d). The total outgoing sediment flux and point denudation with time are correctly reproduced (Figs. 5e and f), though also a little higher than in the true model, because of the slight overestimation of the uplift rate.

### 3.1.4 Natural cases

*- SE Border of the French Massif Central*
The first application to natural cases consists in reconstructing the topography of the escarpment located along the SE border of the French Massif Central. The uplift of the Massif Central occurred in several steps since the Paleogene, the last stage being a Late Miocene to Pliocene uplift which resulted into intense fluvial incision observed along this escarpment (Olivetti et al., 2016; Fauquette et al., 2020). Denudation rates in the recent period range from 0.04 to $0.08 \ \mathrm{mm.yr^{-1}}$ and morphometric analyses indicate a strong asymmetry between agressor catchments located on the steep face of the escarpment, and victims on the opposite side, i.e. draining towards the NW (Olivetti et al., 2016). In the inversion procedure, we cannot accurately constrain time variations of the erosion parameters, which trade off with model duration (the largest the erodibility coefficient, the shortest the time needed to achieve a given amount of erosion). To address this ambiguity, we adjusted the erosion parameters to obtain denudation rates consistent with present-day estimates (Olivetti et al., 2016) as well as with long-term exhumation rates over several million years (Olivetti et al., 2020). We do not aim, as well, at reproducing specific morpho-tectonic events like the Messinian Salinity Crisis, which occurred at the end of the Miocene, and we neglect the — presumably small — topographic uplift that could result from flexural isostatic response to ongoing erosion during the model duration. Consequently, the initial topography that we will obtain from the adjoint method gives a first-order image of the pre-incision landscape, but its precise age remains unconstrained.

The initial guess for the topography is either (1) a NE-SW trending divide, featuring a gently NW-dipping plane opposite to a steeper SE-dipping surface, or (2) the current topography. In the first case, the maximum altitude is 2000 m, located along the drainage divide, while in the second case, it reaches only 1500 m in the NW corner of the model, which corresponds to the Miocene Cantal stratovolcano. The model runs for 3 Myrs with a diffusion coefficient of $10^{-2} \ \mathrm{m^2.yr^{-1}}$ and an erodibility coefficient of $10^{-6} \ \mathrm{yr^{-1}}$. In this inversion, only the final topography serves as a reference observable to compute the functional

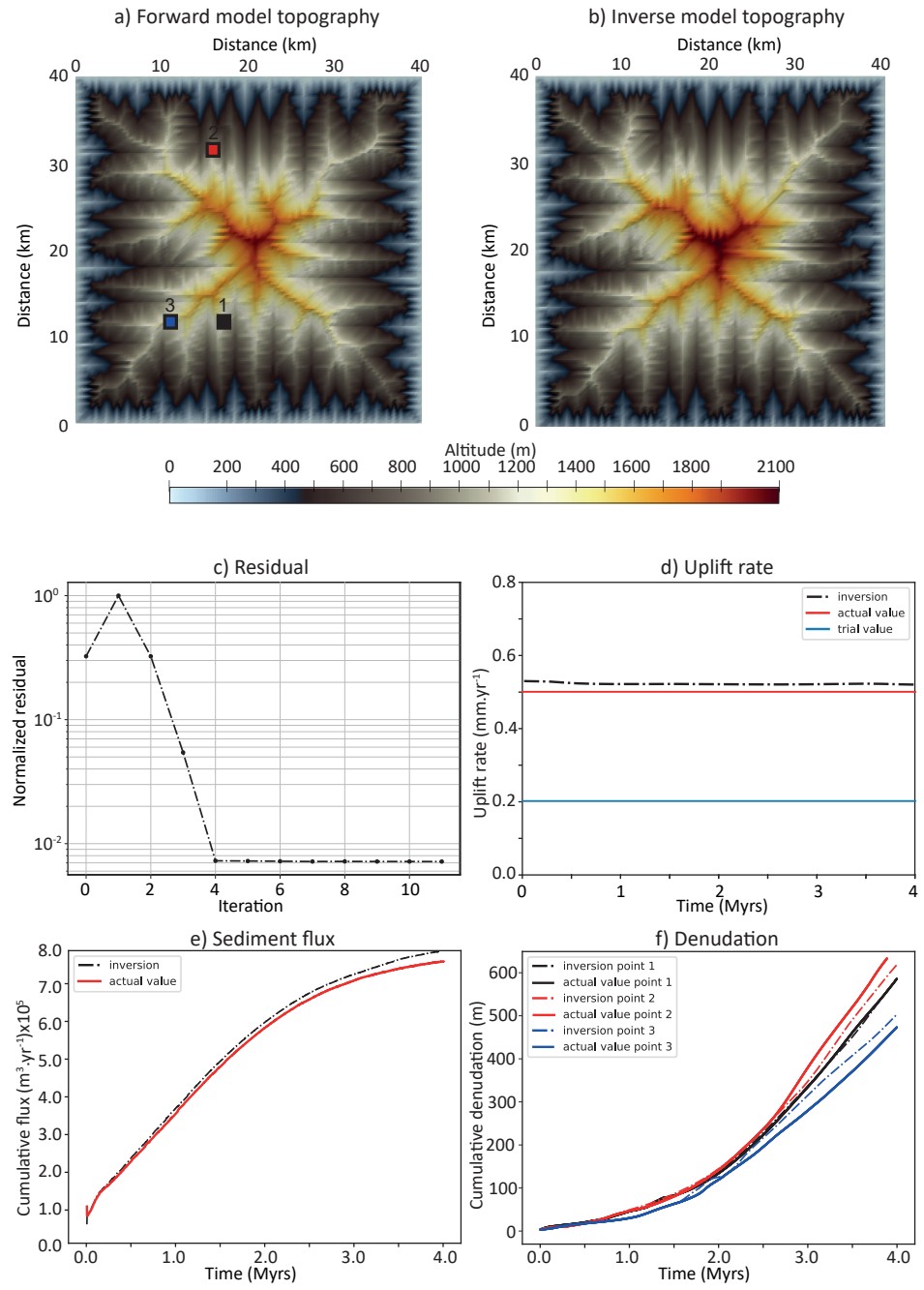

**Figure 5.** Results of source term (constant uplift rate) inversion (model 6, Table 1). Top: final topography with the actual (a) and inverted (b) uplift rates after 4 Myrs. Middle: normalized functional value (c) and uplift rate (d). Blue line on panel d indicates the trial uplift rate value used in the inversion procedure. (e): outgoing sediment flux from the forward ("actual value") and inverse model ("inversion"). (f): actual and modelled cumulative denudation on the 3 control points (see (a) for point location).

$J$. For both initial guesses (divide or current topography), the inverted initial topography reveals a relatively smooth, poorly dissected region in the NW half of the model, representing the footwall of the normal fault system, and a lower-altitude plain in the SE half. In both cases as well, these surfaces are delineated by a linear, well-marked escarpment.

When the initial guess features a smooth topography with a linear drainage divide, the fault escarpment closely follows the observed surface fault traces, and the average footwall elevation is higher than in the second case. In contrast, the model initialized with the present-day topography produces lower elevation contrasts and a slightly less sinuous escarpment, located farther inland relative to the fault trace. Moreover, the deep incision of west-flowing rivers across the Grands Causses area is less efficiently removed than in the simulation using a linear drainage divide as the initial condition. These results suggest that using the current topography as the initial guess is less effective in reconstructing the pre-incision escarpment morphology, leading to a slightly lower mean elevation and a reduced ability to reverse fluvial incision compared to the model starting from a simple linear NE–SW divide.

Geologically, the reconstructed initial conditions from the "divide" model reveal a contrasting landscape within the Cévennes fault system footwall. In the SW, the topography is characterized by a low-altitude (800 m) domain corresponding to the Mesozoic Great Causses and the Variscan Montagne Noire metamorphic massif. To the north lies a moderately elevated region (1000-1400 m), encompassing the crystalline core of the Massif Central. This area features outcropping Variscan metamorphic rocks and granitoids locally overlain by Cenozoic volcanism. At the northeastern edge of the domain, a large circular massif corresponding to the cenozoic Cantal stratovolcano reaches altitudes exceeding 1400 m, however significantly lower than the 2500 m estimated from pollen assemblages prior to its Plio-Pleistocene erosion (Fauquette et al., 2020). In this model, the incision along the southeastern border of the Massif Central by steep, southeastward-flowing streams is largely erased, with only a few notches remaining downstream. The base of the morphological escarpment is well aligned with the main Cévennes Fault, as well as with the lithological boundary between the crystalline basement of the Massif Central and the sedimentary cover of the Rhône Plain. This boundary also coincides with a major early Mesozoic normal fault. These observations suggest that, prior to incision, the morphological escarpment along the southeastern border of the Massif Central was not purely tectonic (i.e., a vertical step resulting from the Mesozoic Cévennes fault system activity), but also reflected a lithological contrast between crystalline and sedimentary units produced by the vertical displacements along this normal fault system.

*- Uplift rate of the Wasatch Fault, Utah*

The Wasatch Fault, in the NE part of the Basin and Range is a well-known active normal fault, having been the focus of numerous tectonic and geomorphological studies since the 1990s. While long-term ($\approx$10Myrs) denudation is estimated to have fluctuated between 0.8 and 1.2 $\mathrm{mm.yr^{-1}}$ (Ehlers et al., 2003), present-day estimates yield much lower values, around 0.2 $\mathrm{mm.yr^{-1}}$ (Stock et al., 2009). Locally uplifted fluvial cave sediments yielded uplift rates around 0.2 to 0.3 $\mathrm{mm.yr^{-1}}$ in the Pleistocene followed by an acceleration to 1.2 $\mathrm{mm.yr^{-1}}$ in the Holocene (Mayo et al., 2009), but at the scale of the fault system comparison between geodetic and geologic fault throw rates evidences large variations that can be due to the superimposed effect of far-field tectonics, earthquake cycle and local fault dynamics (Friedrich et al., 2003). Overall, despite the significant amount of data on this fault system, there is no consensus on the long-term uplift rate of the footwall and its evolution through time. In this model, we treat the source term (uplift rate) as an unknown parameter and we only use the final topography to

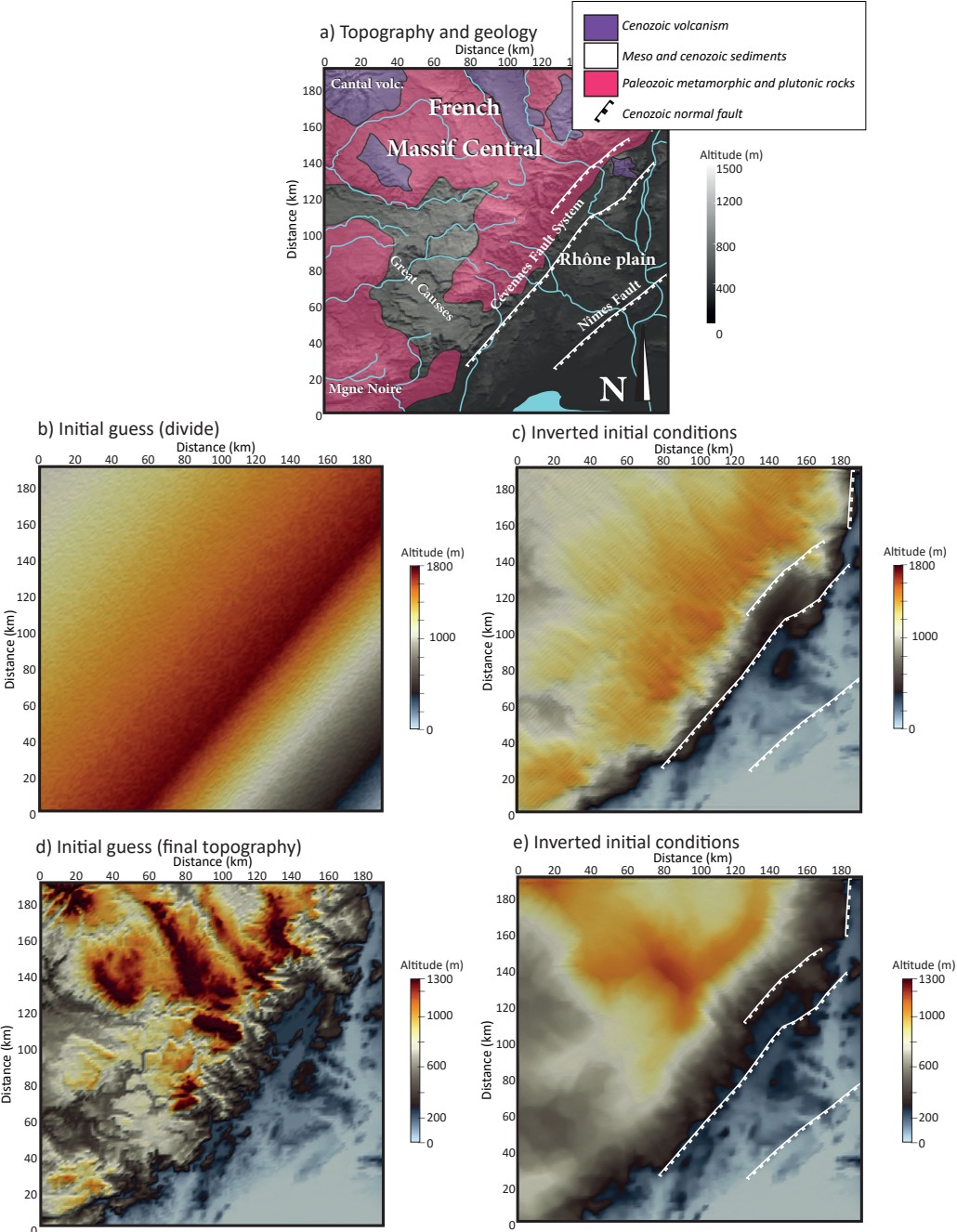

**Figure 6.** Inversion of initial conditions for the French Massif Central (model 7, Table 1). (a): Simplified geology (transparent colors) and present-day topography (grey shades) of the SE Massif Central with the main cenozoic faults (white lines). (b): guess initial conditions with a NE-SW asymmetric divide; (c): inverted initial conditions with (b) as guess; (d): guess initial conditions corresponding to the current topography; (e): inverted initial conditions with (d) as guess.

compute the functional $J$. The target topography corresponds to a $\approx 14$ km-long section of the NS-trending Weber segment of the Wasatch Fault, which separates the Wasatch Range to the east from the Great Salt Lake basin to the west (Fig. 7a).
The initial topography reflects the general shape of the Wasatch Range in this area, featuring a NNW-SSE trending ridge with an elevation of 2500 m that overlooks a plain at an elevation of 1300 m (Fig. 7b). The model simulates 4 million years of landscape evolution (Fig. 7c), capturing the time frame during which river incision may still preserve a record of recent uplift history (Smith et al., 2024). We set the erosion parameters as follows: the diffusion coefficient $\kappa$ is $0.1$ m$^2$.yr$^{-1}$, and the erodibility coefficient $K_f$ is $3.2 \times 10^{-6}$ yr$^{-1}$, close to the values used in previous inverse modeling studies by Smith et al. (2024). The initial guess for the uplift rate is $0.5$ mm.yr$^{-1}$. The residual drops significantly within the first 10 iterations, and the model stops after 24 iterations (Fig. 7d). The estimated uplift rate shows an initial phase with relatively low values (less than $0.2$ mm.yr$^{-1}$), gradually increasing to approximately 1 mm.yr$^{-1}$ over 4 million years of evolution. This result aligns with the average Quaternary uplift rates obtained from river longitudinal profile inversion (Smith et al., 2024) (Fig. 7e) especially between 2 and 3.2 Myrs of model evolution where the mean uplift rate seems to increase regularly. In the last 1 Myrs, river profiles tend to indicate a drop in uplift rate values from $0.8$ to $0.5$ mm.yr$^{-1}$ while our inversion still predicts an increase, that is consistent with some short-term uplift rate estimates (Mayo et al., 2009) (Fig. 7e), but not with all. By dividing the modeled outgoing sediment flux by the area of the uplifted region, we estimate average denudation rates ranging from $0.2$ to $0.35$ mm.yr$^{-1}$ (Fig. 7f), consistent with values derived from cosmogenic $^{10}$Be measurements (Stock et al., 2009).

## 4  Discussion

These examples highlight the potential of solving a diffusion-advection equation for both forward and adjoint problems in landscape evolution. Notably, the ability to compute model sensitivity to key parameters such as erosion rates, source terms, or initial conditions provides important insights into the dominant drivers of landscape evolution in both generic and case-specific scenarios. Furthermore, this approach widens the scope of data-driven inverse modeling by enabling the integration of diverse constraints, including time-dependent (e.g., denudation rates, sediment flux) and time-independent (e.g., final topography) observations into the inversion and to invert for a large number of parameters like the initial topography, or spatial variations $\kappa$ or $K_f$.

For comparison, Yuan et al. (2019) conducted 100,000 forward model runs to constrain four spatially constant erosion parameters two of which define the minimum and maximum bounds of a temporally evolving fluvial incision parameter $K_f$ (which varies linearly over time). Pedersen et al. (2018) performed approximately 13,000 forward model runs to constrain between 2 and 5 free parameters related to uplift rate variations and erosion coefficient, the uplift parameters being defined within a prescribed spatial uplift zone and allowed to vary only across discrete time intervals. While such methods can identify the best-fit model at a reasonable computational cost, they lack the capacity to precisely quantify spatial sensitivity, that is, they cannot determine which specific topographic features most strongly influence the fit to observations.

The adjoint method offers clear advantages for inverse modelling in landscape evolution. It provides an efficient and rigorous means of computing the sensitivity of a misfit function to all model parameters simultaneously, regardless of their number. This

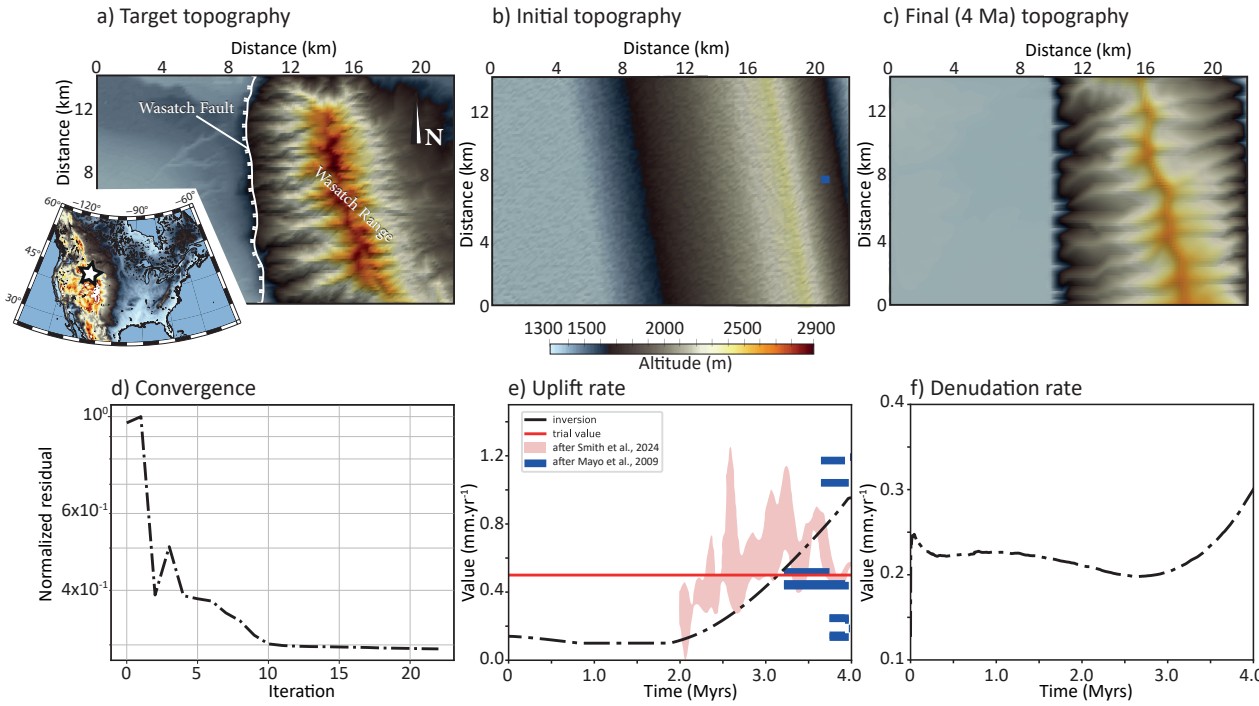

**Figure 7.** Inverse modelling of the Wasatch Range uplift rate (model 8, Table 1). (a): Topography of the Weber segment of the Wasatch fault used as the control parameter (white star on inset indicates its location in North America); (b): Initial topography conditions for the inverse model; (c): Modelled topography with uplift rates obtained by the inverse model; (d): model residuals; (e): Comparison between inverted uplift rates (this study) with those obtained from river longitudinal profile analysis (Smith et al., 2024), red enveloppe corresponding to 7 profiles, and from absolute dating of fluvial sediments in a cave (Mayo et al., 2009); (f): modelled denudation rate.

allows for a detailed understanding of how individual parameters influence model outcomes, without the need for extensive sampling of the parameter space through thousands of forward model runs, as required by gradient-free methods. Additionally, while incorporating more complex processes (e.g., sedimentation, isostatic rebound) significantly increases computational costs for gradient-free methods, the adjoint method remains computationally feasible as the forward model is only run once.

While solving erosion as a PDE with the adjoint method offers significant advantages, such as efficient gradient computation and sensitivity analysis, it has several limitations. In our case, when advection velocities become high, the drainage network can become numerically unstable, leading to hazardous propagation of local extrema. This issue stems from solving the stream power law as an advection equation rather than being specific to the adjoint method itself. A second constraint lies in the need for accurate, independent computation of drainage areas. The method requires that drainage directions align as closely as

possible with the true topographic gradient, since upstream information propagation along channel networks depends on this alignment.

Another limitation of the adjoint approach is that it relies on gradient information, which means it is most effective when the initial parameter guess is reasonably close to the true solution. If the starting point is too far from the optimum, the method may converge slowly, get trapped in local minima, or fail to find a meaningful solution. In contrast, gradient-free methods, such as neighbourhood algorithms or Bayesian sampling, can explore the parameter space more globally without requiring a good initial guess, although at a higher computational cost.

Currently, our approach remains restricted to detachment-limited erosion scenarios where the SPL applies and where the slope exponent $n = 1$. Significant development work would be needed to extend this framework to more complex situations, such as transport-limited systems involving sediment deposition or flexural isostatic adjustment. Furthermore, incorporating additional constraints like cosmogenic exposure ages would enhance the method's applicability but still has to be implemented.

In this study, we have limited the model to simple inversions under two main assumptions: 1) only one type of parameter (even though spatially variable diffusivity or initial conditions involve as many parameters as there are grid nodes) is unknown, the other ones being perfectly determined; and 2) while the model can handle spatially variable erosion coefficients and initial topography, it cannot determine both temporal and spatial uplift rate variations. While such conditions are not always realistic in natural settings, this limitation is not unique to inverse problems; forward models of landscape evolution also face similar constraints. Like in forward models, another major tradeoff could arise from the under-determination of both the initial conditions and the uplift rate, or erosion coefficients and uplift rate in the case of an actively uplifting landscape, especially if the steady-state is not yet reached. For these reasons, joint inversion of multiple parameters such as the diffusion and erodibility coefficients, as well as source and initial conditions, could improve the procedure. The challenge, however, consists in incorporating sufficient data so that the joint inversion problem is not overly under-constrained.

Applying this method to natural cases demonstrates its potential to infer pre-incision topography or temporal variations in uplift rates, provided reasonable assumptions are made about the remaining parameters. In the case of the French Massif Central, while the age of the pre-incision topography remains uncertain, the recovered initial topography effectively highlights a well-defined linear escarpment. This inversion proves relatively straightforward because the original fault escarpment has not been extensively degraded. Moreover, model sensitivity estimates highlight a domain with a large sensitivity both to diffusion and erodibility coefficients. This domain, located along the escarpment of the Mesozoic Cevennes fault system is presently characterized by intense seasonal rains called "Cevenol episodes" due to the conjonction of a topographic barrier to southerly winds carrying the humid and warm air of the mediterranean Sea (e.g., Delrieu et al., 2005). Assuming that the erodibility coefficient $K_f$ is directly related to the precipitation rate, we can deduce from the sensitivity maps that the Cévennes landscape evolution could be very sensitive to these extreme episodes. Finally, the inversion of temporal variations in uplift rates along a segment of the Wasatch Fault is feasible due to the strong dependency of the topography — particularly river longitudinal profiles — on the uplift rate. However, as regressive erosion progressively removes evidence of the earliest landscape history, it is virtually impossible to reconstruct the oldest phases of uplift. The observed discrepancies between uplift rates derived from river profile inversion and our adjoint-based approach likely stem from several key factors. First, river profile methods rely on a steady-state assumption linking channel morphology to uplift history (Goren, 2016), an approximation that may not always be valid. Second, such approaches inherently focus on channel dynamics while neglecting hillslope processes and the rest of the

landscape. Third, they typically assume temporally invariant drainage areas, ignoring catchment reorganization with time. In contrast, our adjoint method uses a reduced functional that minimizes the residual between modeled and observed topography, ensuring consistency with the range-scale morphological evolution. While this approach better captures the average landscape response, it naturally reduces the importance of second-order features like knickpoints that record abrupt uplift changes.

## 5 Conclusion

A finite-element model that combines the Stream Power Law of fluvial incision with linear hillslope diffusion into a classic diffusion-advection equation offers several advantages. First, it enables a consistent treatment of the "slope" responsible for both the downslope movement of the surficial altered layer (simulating hillslope diffusion) and water flow along channels (fluvial incision). In this approach, both processes rely on the same topographic gradient, providing a more physically realistic framework than the classical models that use a local channel slope derived from preferred drainage directions. Second, it enables the solution of the adjoint problem to invert unknown parameter values and assess the model sensitivity to these parameters, which is difficult to do with only forward models and gradient-free procedures.

Our results demonstrate that it is possible to invert spatial variations of the diffusion and erodibility coefficients, and to evaluate the landscape sensitivity to these parameters. Additionally, simple uplift scenarios involving constant or smoothly varying rock uplift rates can be effectively determined. When addressing the initial conditions of a degrading escarpment, some regularization parameters are required, yet this approach still yields consistent results with respect to the forward model.

Applications to real-world data, such as the pre-incision topography of the SE French Massif Central and the uplift rate of the Wasatch Fault in Utah, suggest that this methodology holds potential for natural case studies. In particular, we show that the dissected, linear escarpment bounding the SE part of the French Massif Central is more sensitive to hillslope diffusion and river incision than the rest of the domain. In the same region, our results suggest that, prior to intense incision by SE flowing rivers, the topography was delineated by a linear NE-SW escarpment that coincides with the trace of a major Mesozoic fault. Concerning the Wasatch fault, slip rate inversion depicts a first period of low slip rate, that has increased dramatically 2 Myrs ago.

*Acknowledgements.* Many thanks to Gareth Roberts, Guillaume Duclaux and Tristan Salles for very enthousiastic and fruitful discussions on this paper. We thank both reviewers John Armitage and Stefan Hergarten for their insightful comments, and Associate Editor Fiona Clubb for her constructing remarks on the second version of this paper.

Supplementary material: https://doi.org/10.5194/esurf-0-1-2025-supplement

*Author contributions.* CP developed the code, analyzed the results and wrote the manuscript. AJ developed the code, discussed the results and participated in writing the manuscript. NC discussed the results and participated in writing the manuscript.

*Code availability.* The code, parameter and data files are available on bitbucket (git@bitbucket.org:jourdon_anthony/laser.git).

## Appendix A: Incompressibility of the Stream Power Law

To demonstrate the incompressibility of the Stream Power Law (SPL) assuming $n = 1$, let us consider the sediment flux

$$\mathbf{Q_f} := \mathbf{c}h \tag{A1}$$

as the product of a velocity vector $\mathbf{c}$ and the scalar quantity $h$ representing the topography. The conservation of the transport of topography is given by the continuity equation:

$$\frac{\partial h}{\partial t} + \nabla \cdot \mathbf{Q_f} = U, \tag{A2}$$

where $U$ is the source term (uplift rate). Substituting Eq. (A1) into the continuity equation and applying the product rule yields:

$$\frac{\partial h}{\partial t} + \mathbf{c} \cdot \nabla h + h \nabla \cdot \mathbf{c} = U. \tag{A3}$$

Using the definition of $\mathbf{c}$ given by Eq. (6), Eq. (A3) becomes:

$$\frac{\partial h}{\partial t} + K_f A^m \mathbf{u} \cdot \nabla h + h \nabla \cdot (K_f A^m \mathbf{u}) = U. \tag{A4}$$

Noting that by the definition of $\mathbf{u}$ given in Eq. (7):

$$\mathbf{u} \cdot \nabla h = \frac{\nabla h}{\|\nabla h\|} \cdot \nabla h = \frac{\|\nabla h\|^2}{\|\nabla h\|} = \|\nabla h\|,$$

we recognize that

$$K_f A^m \mathbf{u} \cdot \nabla h = K_f A^m \|\nabla h\|,$$

is the transport term of the SPL for $n = 1$. Therefore, considering that the SPL can be seen as a kinematic wave equation describing the transport of topographic information, the continuity equation becomes

$$\frac{\partial h}{\partial t} + K_f A^m \mathbf{u} \cdot \nabla h = U, \tag{A5}$$

which implies that $\nabla \cdot \mathbf{c} = 0$ and demonstrates that the use of the SPL as a transport equation for the topography assumes the incompressibility of the rate at which the topographic information is transported.

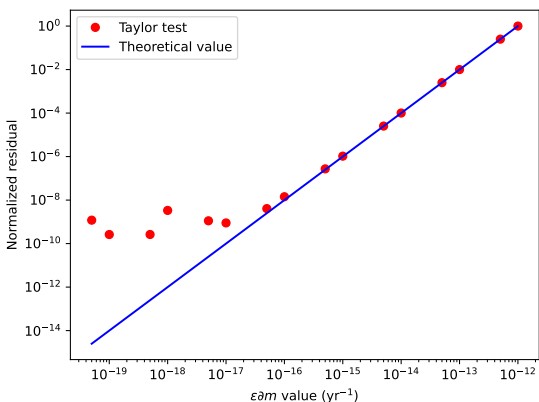

**Figure B1.** Taylor test on the erodibility coefficient $K_f$ using an initial $\epsilon$ value of $1 \times 10^{-6}$ (upper right point) with a constant $\partial m$ value equal to $1 \times 10^{-6} \mathrm{yr}^{-1}$. Model 9, Table 1.

### Appendix B: Taylor test of the adjoint model

We test the consistency of the adjoint model using a Taylor test on an initial forward model consisting of a $40 \mathrm{~km} \times 40 \mathrm{~km}$ domain, with a $20 \mathrm{~km} \times 20 \mathrm{~km}$ square mountain at the center uplifting at a constant rate of $0.5 \mathrm{~mm.yr}^{-1}$. The Taylor test verifies that:

$$J(m + \epsilon \partial m) - J(m) - \epsilon \nabla J \cdot \partial m \to 0 \quad \text{at} \quad \mathcal{O}(\epsilon^2), \tag{A1}$$

where $m$ is the control parameter of interest, $\partial m$ is a small perturbation of this parameter, and $\epsilon$ is a small value. The Taylor test consists of applying a perturbation to the initial erodibility coefficient $K_f$, computing the second-order Taylor remainder using Eq. (A1), then successively decreasing $\epsilon$ and verifying that the residual decreases at the expected order of $\epsilon^2$.

The results of the Taylor test applied to this model show that the residuals are consistent with theoretical predictions (Fig. B1), exhibiting a convergence rate close to 2 down to $\epsilon$ values of $\sim 1 \times 10^{-16}$. Beyond this threshold, the convergence deteriorates, and the residual fluctuates between $10^{-10}$ and $10^{-8}$. This result indicates that the adjoint model is robust down to machine precision (Fig. B1).

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
