# Peer review of "Reconstructing landscapes: an adjoint model of the Stream Power and diffusion erosion equation"

_EGUsphere, 2025_

## Author Response (AR1)

REF1

In this manuscript the authors reformulate the much-used stream power model with additional diffusion term as an advection-diffusion equation and then apply the adjoint method for parameter inversion. The approach is quite novel, and I am broadly supportive of it. I think this manuscript would benefit from a bit more pedagogic explanation of the implementation of the approach and how the sensitivity analysis functions or what it means. I found the two applications of the model a bit limited and without a real scientific question behind them. This left me thinking if they are the best examples to demonstrate the potential applications of the new adjoint method.

**Re:** Thank you for your overall positive evaluation of our work. You are correct that the two examples presented in the manuscript were chosen primarily to illustrate the potential of the proposed method. We believe that, as a first step, it is important to test the method on relatively simple and well-understood examples before applying it to more complex scenarios, which may require additional developments (as explained in the Discussion section). We now better present the interest of these two natural examples in the revised version of the paper by addressing two scientific questions: what was the shape of the SE escarpment of the Massif Central before incision **(lines 386-391)**? How did the Wasatch fault throw rate evolve in the last 4 Myrs **(lines 432-438)**?

In detail my main comments are:

1. The explanation of the adjoint method is a bit messy. The derivation of the weak formulation of the advection diffusion equation is more appropriate for a textbook, while other details are missing, such as how the Dinf routing is used to match the upstream area to a vector normal to the gradient, and how accurate is it?

   **Re:** Considering the remarks of Referee 2, it is essential to justify why we consider the flux divergence term negligible in our formulation. However, instead of discussing this using the weak form of the equation (which indeed was not necessary to detail), we just explain it with the strong form that already implies div(c) = 0. This is corrected in the revised version of the manuscript **(lines 113-145)**.

   Next, we agree that the section currently lacks explanation regarding the use of the D-infinity (D∞) routing algorithm. Our reasoning is as follows: in the Stream Power Law (SPL), the "local slope", often denoted as *S*, typically corresponds to the drainage direction and should ideally be aligned with the topographic gradient. The D8 algorithm provides only a crude, single-direction estimate of flow direction, whereas the D∞ method offers a more accurate representation by distributing flow among multiple downslope neighbors, thus yielding a direction that better approximates the true gradient. Moreover, it permits divergent drainage, which can happen in reality but is impossible to reproduce with D8 (Tarboton, 1997) **(lines 148-153)**.

   We compute the drainage area using the D∞ method, and then construct an advection velocity field **c**. These vectors have a magnitude equal to $K_f A^m$, and a direction aligned with the topographic gradient. This construction allows us to express the advection term in a way that is consistent with both the physical intuition of the SPL and the mathematical

structure of the advection-diffusion equation (see response to reviewer 2). We have added some explanations in the revised manuscript (**lines 190-197**).

To clarify the discussion, we present here (in the pdf file) a figure was meant to be in the paper at the origin, but was finally removed because it is not strictly necessary. It presents 3 different ways of computing drainage and erosion. A: drainage computed with the D8 method, erosion computed with the classical finite-differences SPL + diffusion; B: same as A, but the drainage is computed with D∞; C: same as B, but erosion is computed with the diffusion-advection equation (our approach). In all plots, red vectors have a norm equal to $K_f A^m$ and trend parallel to the topographic gradient. On A, the drainage network is not parallel to the topographic gradient and large erosion rates (large vectors) occur only in the main channels; in B, this effect is less important but there is still a discrepancy between the drainage direction and the topographic gradient. In C, there is a better adequation between the drainage network and velocity vector directions, and a less strongly localized incision.

We have modified the methodology part of the paper in order to account for this remark and better justify the use of the D∞ routing algorithm for our purposes **(lines 190-197)**.

[Figure]

[Figure]

[Figure]

2. The model relative sensitivity is a bit obscure to me. I am used to sensitivity analysis requiring a range of parameters to be chosen, such as running the model for a range of Kf, kappa, m, etc. The sensitivity is therefore a function of both the range of potential values of each parameter and the model response. Here however, the parameters are fixed (line 196). I think some clearer explanation of what the authors mean by sensitivity is required, because I am confused.

**Re:** In this study, the term sensitivity refers to the **local sensitivity** of the model output (more precisely, the cost functional) with respect to small perturbations in model parameters. This is distinct from global sensitivity analysis techniques, which evaluate model response over a broad range of parameter values. Using the adjoint method, we compute the gradient of the cost functional with respect to each model parameter at a fixed reference point in parameter space (i.e., a given model run), which provides the relative influence of each parameter on the model output **at that point**. This approach does not require varying parameters across a range, as in the Morris method, and is computationally efficient even in high-dimensional settings. However, it provides *local* information only, which means that sensitivities are valid in the vicinity of the tested parameter set. We use this to construct relative sensitivity maps that highlight areas of

the domain where the model is most responsive to changes in specific parameters, such as Kf or κ. Since the parameters differ in units and magnitude, comparing their absolute sensitivities directly is complicated. Therefore, we normalize the sensitivities, scaling them between 0 and 1, to allow for comparisons across parameters and visualize their spatial variations. We add a little more explanation on the meaning of model sensitivity in the revised version **(lines 64-75, 85-88, 242-247 and 473-475)**.

3. The misfit in the test with known solutions are never quantified, other than the objective function J. Rather than a visual description of how well the inversion does, it would be good to quantify the misfit. This could lead to the potential to consider other objective functions that might improve the model fit but perhaps with the loss of some other quality of fit.

**Re:** The misfit is quantified through the objective functional *J*, which measures the squared difference between the model prediction and the reference data (L2-norm). The decrease of *J* over iterations directly reflects the improvement of the inversion. We acknowledge that alternative objective functionals could potentially emphasize different aspects of the model fit, but our goal here was to keep the misfit definition as general as possible for a first methodological study. Exploring alternative formulations of the objective function could indeed be an interesting direction for future work, particularly when targeting specific applications or types of data.

4. The application to the S.E. boarder of the Massif Central is a bit opaque to me. There are quite a few caveats to the application of the model to this location. Would it not be better to apply the model to a landscape where the influence of erosion is a bit "cleaner"? Perhaps somewhere that has been impacted by dynamic uplift, so the ancient landscape is still partially visible?

**Re:** As mentioned in our first response, we intentionally selected relatively simple test cases where only one parameter (such as the initial condition or uplift rate) needs to be inverted. This choice represents a compromise between too simple landscapes—where reconstructing the pre-incision topography would be straightforward and the adjoint method unnecessary—and too complex settings that would require joint inversion of multiple parameters, which is beyond the scope of the present study. While the southeastern border of the Massif Central may experience some uplift, it is relatively modest. Additionally, the landscape is sufficiently dissected, making it a suitable candidate for inversion of initial conditions, but it is still sufficiently preserved to identify where the original escarpment could have been. We have included these justifications in the revised paper **(lines 310-312 and 375-378)**.

Below are point by point questions/comments in the order that they come in the text:

Introduction: I think it would be good to also discuss the many models that invert river long profiles for uplift and past climate. Furthermore, it would be worth discussing the timescale of interest. The study by Barnhart is a sensitivity analysis and inversion for modelling processes on shorter timescales than the intended study in this paper. Have there been any sensitivity analysis of the simpler LEMs as modelled by Equation 1? In the discussion some papers are cited that ran 10's of thousands of models to fit erosion parameters.

**Re:** We agree that the Introduction would benefit from additional references, particularly regarding models that invert river longitudinal profiles for uplift and past climate, such as the work by N. White and G. Roberts, that we forgot to cite **(lines 46-55)**. After reviewing the literature, we found that explicit discussions of model sensitivity in simpler landscape evolution models (as represented by Equation 1) are relatively rare. Nonetheless, we have extracted relevant information on this topic from the literature and incorporated it into the revised Introduction together with the missing references **(lines 65-75)**.

Line 43: I think it is too soon in the introduction to discuss "gradient-free methods" without first explaining the past sensitivity analysis that have been done with qualitative methods such as the Morris Method.

Re: Agreed. Thank you for the references to the Morris method. This is indeed interesting to precise, as this method rests on one-at-a-time modification of model parameters, hence again on a large number of forward models **(lines 65-75)**.

Line 62: I would not cite Simpson & Schlunegger (2003) as they solve a diffusion equation for sediment transport, not the advection-diffusion equation described in equation 2.

Re: Thanks for this comment, we have removed this reference here.

Line 66: This explanation is confusing. "u" is a unit vector in the direction of drainage, while its magnitude is $K_f A^m$, where A is a function of the position. I am not sure if the magnitude of this vector corresponds to the speed of knick-point migration, so the analogy might not be useful.

**Re:** To clarify, **u** is a unit vector representing the drainage direction. When multiplied by the scalar $K_f A^m$, it yields the advection velocity vector **c**, whose magnitude varies spatially depending on both the drainage area A and the parameter $K_f$. We realize this was not clearly stated in the original manuscript, so we will improve the explanation in the revised version. Regarding the analogy with the knickpoint migration rate, it may have been oversimplified or inadequately explained. As noted by many authors (i.e., Whittaker and Boulton, 2008), solutions to the stream power erosion law can be described as nonlinear kinematic waves, with an intrinsic wave celerity representing the knickpoint retreat rate. In the classical stream power law framework, this velocity is given by:

$$C_E = \psi A^m S^{n-1}$$

which simplifies to:

$$C_E = \psi A^m$$

when n=1. Here, $\psi$ represents the bedrock's sensitivity to erosion, similar to the erodibility coefficient $K_f$.

In our definition, the advection velocity **c** describes how topographic information propagates upstream, encompassing not just knickpoints but all topographic information: we treat any topographic variation as a physical quantity moving upstream at velocity **c**.

Equation 3: How is the direction of u calculated from the $D_{inf}$ routing algorithm?

**Re:** The direction of **u** is not computed from the D∞ routing algorithm. Instead, it is derived directly from the gradient of the topographic grid, which we calculate using built-in functions available in Firedrake such as the shape functions derivatives. This approach ensures that **u** accurately reflects the local topographic slope direction **(lines 125-140).**

Line 88: It would be good to explain this a bit further: why would it be "physically meaningless"?

**Re:** As explained in the manuscript, there is no physical basis for differentiating between the true topographic gradient, which appears in the diffusive term and represents the direction of hillslope sediment transport, and the flow direction of water (often referred to as the "local slope" in the classical SPL formulation). In natural landscapes, both water and sediment flow downhill along the direction of the steepest slope, i.e., the topographic gradient. Therefore, using a "local channel slope" distinct from the topographic gradient in the SPL formulation lacks physical justification, which is why we describe it as "physically meaningless." Maybe the term was not correct, so we have reformulated this explanation in order to clarify it **(lines 162-163)**.

Line 96: What is "g"? There are a few "g"s in this manuscript. I assume it is a fixed topography at all the boundaries, but is this a good boundary condition?

**Re:** Thank you for pointing this out. Indeed, g represents the prescribed boundary condition for h on the model boundaries. While it is a fixed boundary condition in the sense that it is imposed at all times t∈[0,T], g can vary with time. Specifically, g corresponds to the initial topography plus the cumulative uplift over time at the boundaries. This formulation prevents the model from maintaining an artificially low baselevel in the uplifting regions, ensuring a more physically realistic representation of boundary conditions **(lines 201-202)**.

Lines 103 to 110: This paragraph is a bit of a clumsy mix of supplementary information. Perhaps it should be redistributed into the rest of the model description.

**Re:** Thank you for your suggestion, we have tried to better organize the model description in the revised manuscript **(new sections 2.1 and 2.2)**.

Line 115: I understand that "c" is a variable, but it is not like the others, as it is a function of the evolution of the model. Can it be called a parameter? I am guessing this is not a big problem, but perhaps some clarification for non-experts in the adjoint method (people like me) would be useful.

**Re:** The distinction here is largely semantic. Based on what we have read, a *parameter* is fixed or prescribed by the user, and makes the links between *variables*. The latter are mathematical quantities which can evolve as the model runs and are linked together by a relation. The advection velocity **c** depends on the topography *h* (variable) and drainage area *A* (variable)- both of which evolve with time during the simulation- via the parameters Kf and m. Therefore, it is more appropriate to refer to **c** as a *variable* rather than a *parameter*.

Equation 6: another "g", but this is not the same as the boundary condition "g". Right?

**Re:** Indeed, thanks for pointing it out. There are too many "g"s in the paper. This is being corrected **(section 3.1)**.

Equation 13: The definition if $J_{reg}$ is maybe a bit out of place, or Equation 15 is out of place, as there is a J and a $J_{reg}$ and then in equation 16 some more terms to a new J. I think the description of the cost term could be much better organised.

**Re:** Thank you for this suggestion. We have corrected this part **(section 3.1).**

Section 4.1: I think this would be better as an appendix. It ruins the flow of the text.

**Re:** This section details the Taylor test of the adjoint model, which serves as a fundamental prerequisite when employing the adjoint method. Conducting this test is essential to ensure the correctness of the adjoint. Although we did not apply this test to every model variable, we believe it is important to present this straightforward test within the main body of the paper to justify the subsequent analysis. Additionally, the section is quite brief, which we feel minimizes the disruption to the text's flow. This being said, based on this comment we now consider adding an appendix where technical details like this one could find their place **(lines 560-572)**.

Line 197: I am a bit lost, again likely due to my lack of knowledge in the adjoint method. The sensitivity to the parameters is discussed without varying the parameters. How can this be done? This is not a quantitative or qualitative sensitivity analysis where hundreds to thousands of models are run with different input values the same as this sensitivity analysis. My hunch is that the sensitivity analysis here is not equivalent to that presented by Barnhart et al. (2020). No range of the parameters are tested, so I don't see an uncertainty in the diffusivity, the erodibility, the exponent "m" or the uplift and initial condition.

**Re:** We appreciate your feedback and apologize for any confusion. Since not all readers may be familiar with the adjoint method, we aim to make our paper as clear as possible, even for non-specialists.

The adjoint method enables us to compute the gradient of the cost function *J* with respect to each parameter of interest. This gradient represents the local sensitivity of *J* to changes in those parameters, indicating how a small change in a parameter will affect *J*. Unlike other sensitivity analyses, which involve running multiple models with different input values, the adjoint method provides a more efficient way to obtain these sensitivities by calculating the first-order derivatives of *J*.

This approach is particularly useful because it allows us to determine the sensitivity of the model to each parameter without the need for extensive computational resources. The sensitivity obtained through the adjoint method is local, meaning it depends on the current values of the parameters, typically based on an initial guess (see above).

Line 200: I think instead of "somehow" you mean "somewhat". In any case, better to quantify this difference than to use vague qualitative statements.

**Re**: Thanks for this suggestion.

Figures 2, 3, and 4: I would prefer it if the authors used perceptually uniform colour maps, and even better linear ones, such as "viridis" etc. I am OK with "terrain", but even then, this is not really the best as it draws out specific topographic elevation that have no real significance.

**Re**: Regarding Figures 2, 3, and 4: We agree that linear colormaps are preferable because they prevent the over- or under-representation of specific values. However, representing topographic maps with linear colormaps while maintaining sufficient contrast to highlight both low- and high-elevation areas can be tricky. To address this, we have changed the representation of the topography in all figures by using a divergent linear colormap that is also colorblind-friendly. This approach suppresses non-linearity while providing an adequate distinction between high and low elevations **(all figures but fig. 8 have been modified according to this remark)**.

Figures 3, 4 and discussion in Section 4.1.1: I think that the spatial distribution in the misfit between the inversion and the known distribution of the diffusivity and erodibility could be quantified rather than just explained in the text. Could this not lead to propositions for a better cost function for the adjoint method?

**Re:** We have added a figure illustrating the spatial distribution of the misfit **(new figures 2 and 3)**. However, we maintain that the cost function serves as the most effective quantification of this value. There are already established methods to enhance the cost function, primarily through the adjustment of regularization parameters. These regularization terms are crucial as they allow the inversion process to avoid over-fitting, particularly in areas that are not highly sensitive to parameter variations. By selecting these parameters appropriately, one can guide the inversion process toward either a smoother or a more detailed solution, depending on the desired outcome.

Figure 5: The axis labels are tiny.

**Re**: Thanks, this has been corrected.

Figure 6: For parts (d), (e) and (f) the y-axis label is not very helpful. Part (f) has no label "f", or title.

Re: Thank you for pointing this out, we have corrected that.

Line 261: "peculiar", I think this is not the word the authors would use if they could write this article in French. Peculiar is something that is weirdly odd. Perhaps "specific morpho-tectonic", or "unique"?

**Re:** Thank you for this remark. We have removed this term (a poor English translation of the word "particulier" in French).

Paragraph starting on line 265: How can the authors justify the specific values of erodibility and diffusivity chosen for the inversion? Are these not also free parameters? For the application the Wasatch Fault in Utah the choice of erosion parameters is justified based on previous inversions of river profiles, but here there is none.

**Re:** The erosion coefficients must be fixed prior to the inversion of initial conditions. We need erosion parameters that yield denudation rates consistent with observations over approximately the last 4 million years. Although these parameters are not tightly constrained, they fall within a range that produces consistent results in terms of denudation rates. As an example, if we extract the denudation rates of the final model of the French Massif Central run with the reconstructed initial conditions over the last 1 Myrs, we obtain denudation rates which range from ~80 mm/kyr on average close to the headwaters of the escarpment catchments, to 20 mm/kyr on the smooth surface of the Massif Central. Although our study area is located slightly southward of theirs, these results are completely of the same order as the denudation rates deduced from [10]Be concentration in river sands by Olivetti et al. (2018). This is now explained in the revised manuscript **(lines 393-395)**.

Summary:

I think that this is an interesting contribution the research into inverse modelling of landscape erosion. I however feel like in order to get the most out of this research the authors could explain their method more clearly and explain how sensitivity analysis in the adjoint method compares to either Monte-Carlo methods, or statistical approaches such as the Morris method that have been applied to short-timescale landscape evolution models (e.g. Skinner et al., 2018; Barnhart et al., 2020). I am not convinced by the two test cases, but this is not really a problem as I feel that this manuscript is principally about describing the adjoint formulation. However, a more convincing inversion would be a bonus.

**Re:** We appreciate the reviewer's feedback and acknowledgment of our contribution. We are currently working on additional inversions and sensitivity analyses. For the purposes of this manuscript, however, we would like to focus on these two examples to demonstrate the potential and applicability of our method. Our goal is to provide a clear and accessible description of the adjoint formulation so that other researchers can also begin to utilize this approach. Nevertheless, we have tried to strengthen the scientific questions associated with these 2 examples, in order to present something more interesting to the reader than simple "test cases".

I hope these comments are helpful and useful.

**Re:** They are indeed, thank you.

John Armitage, IFP energies nouvelles
* * *
REF2

This paper is about inverting topography in the context of fluvial erosion and hillslope processes. There have been some papers about this topic in the past years. So I would consider it an important topic in quantitative geomorphology.

I saw that the first reviewer already provided a very detailed review. However, I got stuck much earlier and ended up with three major concerns. Two of these are about the general approach of combining the stream-power incision model with the diffusion equation and the third one about the application of the adjoint method.

(1) At least in the traditional form with discrete flow directions, (D8 algorithm), combining the stream-power incision model with the diffusion equation causes severe scaling problems, which make the results dependent on the spatial resolution of the model (Perron et al. 2008, doi 10.1029/2007JF000977; Pelletier 2010, doi 10.1016/j.geomorph.2010.06.001; Hergarten 2020, doi 10.5194/esurf-8-367-2020; Hergarten & Pietrek 2023, doi 10.5194/esurf-11-741-2023). Obtaining parameter values that depend on the spatial resolution from an inversion would be problematic. At the moment, I do not think that considering the topographic gradient instead of the discrete channel slope, as done here, fixes these problems.

**Re**: This remark is very interesting, and we acknowledge that we may have overlooked this problem. There is indeed a dependency of the results on grid size, as indicated by the cited papers, and as all surface process modelers are aware of. This dependency partly arises from discrete drainage computation, which restricts the width of river systems to the cell size. References to the papers cited above have been added in the revised manuscript.

In our formulation, this problem is much less important because we use a classical advection-diffusion equation in a finite-element scheme, with the only particularity that the advection velocity depends on the drainage area. In our approach, (1) we use the D∞ algorithm to compute the drainage area which is less mesh dependent than the D8 and (2) most importantly we do not use the flow direction predicted by the D8 or D∞ algorithms that indeed depend on the grid, but we use the gradient of the topography computed with the finite element shape functions derivatives. Of course, every numerical method shows some dependency to the resolution used, but using the topographic gradient instead of a simple node-to-node routing attenuates largely the grid dependency as long as the resolution is good enough to capture the first order topography and drainage network **(lines 176-179)**.

(2) Combining the stream-power incision model with the diffusion equation has a weird effect on channel steepness. The steepness index of the channels increases with increasing diffusivity and becomes considerably higher than predicted by the stream-power incision model alone. The problem is that this combination of models describes a downslope sediment flux from the hillslopes (with conservation of volume), but then feeds the transported material into a detachment-limited fluvial model without a sediment balance. Practically, this means that the material brought into the rivers get the same properties (erodibility) as the bedrock. The effect was described in a recent paper by Litwin et al. (2025, doi 10.5194/esurf-13-277-2025), and the authors finally admit already in their abstract that this model combination is unrealistic. The problem was also presented on a conference recently (Hergarten 2025, doi 10.5194/egusphere-egu25-5035). This interference of erodibility and diffusivity would also have a strong effect on the inversion.

**Re:** This is indeed a limitation stemming from the assumption that rocks eroded from hillslopes exhibit the same fluvial erodibility response as the bedrock, which is not always the case. However, our model currently applies to bedrock channels, assuming that the removal of sediments produced from hillslopes is negligible (or very fast) compared to bedrock incision.

Regarding mass balance, it is inherently ensured by the advection-diffusion equation, where the source or sink is represented by the uplift rate U. The diffusive term describes particle motion on the hillslope, and the advective term describes the propagation of topographic information. Implementing a true mass balance would require adding a density multiplier to all terms (with possibly different densities for hillslope sediments), which could be considered in future developments **(lines 144-147)**.

(3) I do not understand the condition div(c) = 0 (incompressibility, line 73) and the "physical" justification given in line 74 does not convince me. I would even suspect that this condition is wrong. For instance, if I assume constant erodibility and m = 1 in a traditional model with discrete flow directions, this condition (div(A) = 0 in this case) would imply that the catchment size does not accumulate downstream, which does not make sense to me. In the general case (div(K_f A^m u) = 0), A and u describe the flow pattern on a given topography, while K_f and m are parameters of the erosion model. The condition would then imply that the catchment sizes A on a given topography depend on the parameters of the erosion model, which is unclear to me. As far as I can see, the entire part of the theory about the adjoint method relies on the assumption div(c) = 0 since the adjoint operator would not be just the original operator with -c instead of c otherwise. Theoretically, even a major part of the theory might collapse if the assumption div(c) = 0 does not hold. However, my concerns may be wrong, but I need a solid argument why div(c) = 0.

Re:

**Re:** This is an important comment, as it raises the question of how the advection velocity **c** should be interpreted and how it relates to the traditional formulation of the Stream Power Law (SPL). We take the opportunity to provide here below a detailed explanation that, we hope, will be useful for the reader.

At its core, the governing equation for surface elevation change due to both hillslope diffusion and fluvial erosion can be written in terms of mass fluxes as:

$$\frac{\partial z}{\partial t} = \nabla \cdot \boldsymbol{Q_h} - \nabla \cdot \boldsymbol{Q_f}. \tag{1}$$

Here, $\boldsymbol{Q_h}$ is the sediment flux coming from the hillslopes and proportional to the topographic gradient (diffusive term), while $\boldsymbol{Q_f}$ is the sediment flux arising from fluvial abrasion. The bedrock detachment (incision) rate $\varepsilon$ is equal to the divergence of fluvial sediment flux $\boldsymbol{Q_f}$ such as $\varepsilon = \nabla \cdot \boldsymbol{Q_f}$.

There are physical and empirical reasons (explained in Perron et al.'s 2008 paper) to relate the incision rate $\varepsilon$ to the drainage area $A$ and to the channel slope $S$, defining the Stream Power Law:

$$\varepsilon = K_f A^m S^n. \tag{2}$$

The local channel slope $S$ is ideally equal to the norm of the topographic gradient:

$$S = |\nabla z|,\tag{3}$$

and we get at the end:

$$\varepsilon = K_f A^m |\nabla z|^n.\tag{4}$$

Moreover, it is a set of physical and empirical parametrizations that permit to express $Q_f$ as a function of $\nabla z$ and make (4) resemble an advective term, with an advection velocity $\mathbf{c}$ such as:

$$|\mathbf{c}| = K_f A^m.\tag{5}$$

What is critical is that in its original formulation, equation 1 can be recast as an advection-diffusion equation **only under the assumption of incompressible flow**, i.e., $\nabla \cdot \mathbf{c} = 0$ and **linear dependency on the topographic gradient**, i.e., $n$=1.

Now, what do we do in our case, is expressing $\varepsilon$ as:

$$\varepsilon = K_f A^m \mathbf{u} \cdot \nabla z,\tag{6}$$

Where $\mathbf{u}$ is a unit vector parallel to the topographic gradient $\nabla z$. So, it is exactly the same as in (4) if we assume $n$=1.

So finally, the equation we solve is:

$$\frac{\partial z}{\partial t} = \nabla \cdot (\kappa \nabla z) - \mathbf{c} \cdot \nabla z,\tag{7}$$

In equation 7, the velocity divergence term is therefore already implicitly null. It resembles a diffusion-advection equation and can be solved as such, but it is not really. It would be, if we were dealing with any physical quantity like heat or any material, but here we are dealing with an information. Here, it is only the shape of the topography that is advected (see seminal papers by Luke 1972 and 1974). **New section 2.1.**

---

## Author Response (AR2)

**Associate editor (Fiona Clubb):**

Many thanks for your constructive engagement with the review process and your response to the reviewers' comments. Your manuscript has now been re-reviewed by both reviewers. Reviewer 1 has some minor comments to be addressed, but Reviewer 2 has suggested there is still a fundamental problem with a core assumption of the adjoint method that you present - that $\nabla \cdot c = 0$. They are concerned that this assumption is at the heart of the method and invalidates the use of the adjoint method. This issue would need to be fully and comprehensively addressed or rebutted in a further response before the paper can be accepted in ESurf.

Re: Thank you for this comment. We agree that this point deserves more thorough explanations, but this assumption is valid under the consideration of the Stream Power Law. Please see the detailed response to Reviewer 2 and the new appendix presented in the revised version of the manuscript.

***Important note to associate editor and reviewers***: *Since the original version of our code that was used for the first version of the paper, the Firedrake package has been upgraded. While we initially intended to maintain compatibility with the original version used in our code, that version has become increasingly difficult to install due to numerous dependency issues. To ensure the reproducibility of our results with minimal installation effort, we have modified and updated our code to be compatible with a more recent release of Firedrake. This update has slightly affected the results of the adjoint model, especially those dependent on regularization parameters. It changes a few results in initial condition inversions, with a more realistic topography in the Massif Central case. The rest is not different from the previous version. We therefore provide updated figures and text describing the results obtained with the new Firedrake version, along with a link to the repository containing the new code. We also took this opportunity to fix the bug about outgoing sedimentary flux raised by reviewer 1, improving both the stability and inversion performance with the updated Firedrake version.*

If you feel like this issue can be addressed and would like to submit a further revised manuscript, please also address the following points in addition to those raised by the reviewers:

Reviewer 1 mentioned in the first round of reviews that the use of the case studies was limited and there were not clear scientific questions behind them. The revised manuscript does a much better job of addressing clearer scientific questions, but I think that the structure is still odd here. None of the method sections contain any information about the purpose or setup of the synthetic or natural case studies, which are also not properly mentioned in the introduction. This makes it difficult for the reader to understand what results are actually presented in the manuscript. I suggest expanding Section 4.1 and moving it to the methods, as well as better integrating the case studies/scientific questions throughout the intro and methods. For example, the purpose of choosing these specific case studies should come earlier in the paper.

Re: Thanks for this suggestion. This has been taken into account. In the revised version the forward model, adjoint and the case studies are embedded in a "Methods" section. We have added a synthetic overview of the scientific questions in the intro, and modified the "natural cases" section in order to avoid redundancy (lines 106-115, 368-378 and 426-431).

Figure 2: panel d is missing an x axis

Re: Thanks for pointing this out. However, we do not see this in our version of the manuscript, maybe something went wrong during the conversion and upload of the paper. We will carefully review the files during the submission process to ensure that everything is as expected.

Figure 3: legend label is cut off in panel c.

Re: Thanks for pointing this out. We do not see it either in our version but we will take care of display issues during the submission process.

**Referee 1 (John Armitage):**

The manuscript is a significant improvement on the original submission and I thank the authors for taking the time to consider my earlier comments. I do however note a few questions/comments that could be addressed:

Line 150 to 156: The Firedrake space function is mentioned without the context of what Firedrake is or does. I think that these technical details could come after the description of the adjoint method and the introduction of the Firedrake package that aids the implementation of the adjoint method.

Re: Thanks for this suggestion. This is now explained in more detail in the revised version of the paper (lines 100-105).

Section 4.1.1: It is possible that I missed it, but it is implicit that m=2 for the tests. If it is not stated in the methods, then it should be. If it is stated in the methods, maybe a short sentence here would be helpful to remind the reader.

Re: in our tests, the $m$ coefficient is always equal to 0.5 but it can be changed by the user. It is true that we omitted to precise this point, and thanks to this remark this error has been corrected (lines 140-143).

Figure 1: Does sensitivity scale linearly with topographic gradient, for both the diffusivity coefficient and the erodability? It would be interesting to see a comparison with gradient as well as topography.

Re: We have made some tests in order to plot the sensitivity to the topographic gradient. The two following graphs show the scaled sensitivity with respect to the magnitude of the topographic gradient for the same model as on Figure 1. Although there seems to be a slight dependency for the erodibility ($K_f$) coefficient, the points are still very scattered, probably because there is also a strong dependency of the fluvial erosion rate on the drainage area. For the diffusion coefficient, it's even less clear, probably because the model is more sensitive to the topographic curvature than to its gradient.

[Figure]

[Figure]

Line 265 and more generally: what is the criteria for convergence of the adjoint method?

Re: convergence of the inverse problem is measured via the reduction of the misfit function. In this case it is a residual variation threshold (around 0.1%) and/or a number of iterations (30 to 50 depending on the cases). This is now explained in the revised version (lines 223-225).

Section 4.1.2: I find the rectangles of different coefficients a little arbitrary. Are they inspired by the sensitivity tests? Furthermore, would there be merit in doing a sort-of checker board test to explore what regions the model can resolve? I'm thinking of something like the classic test that is done in seismic tomography to explore in which regions the inversion can be interpreted.

Re: Thank you for this remark, now we have changed this test in order to incorporate a checkerboard instead of two simple rectangles, which were indeed completely arbitrary. Text has been modified between lines 299 and 328.

Line 287 to 289: The coefficients alpha and k are not chosen arbitrarily? It would be interesting to know the inversion sensitivity to alpha and k (in Equation 19).

Re: There is indeed some necessary tuning to ensure that the coefficients alpha and k are appropriate for the inversion. As in most inversion procedures, regularization plays a crucial role, and the choice of regularization parameters strongly influences the results. A systematic exploration of the effects of alpha and k lies beyond the scope of this paper; however, we agree that readers should have some indication of the expected behavior when varying these parameters. To address this, we have added a graphical table as supplementary material showing several tests that illustrate the impact of changing alpha and k on the residual map (i.e., the integrand of $J$ as defined in Equation 18).

On this graphical table, one can observe that increasing alpha suppresses short-wavelength residual noise, while increasing k promotes the grouping of regions with similar altitudes into a smaller number of broader, nearly flat surfaces. See text lines 245-247 and new supplementary material.

Figure 5c: The residual does not evolve or converge as in the other tests. Something is odd here, I think.

Re: Thank you very much for pointing this out. There was indeed a problem on the way time-distributed controls (like the outgoing sediment flux) were implemented to compute the residual functional, that prevented the residual to decrease correctly. Fortunately, this bug impacted only this part of the model. We have corrected this in the code and provided a new inversion test in the revised version of the manuscript (lines 355-359).

**Referee 2 (Stefan Hergarten):**

Dear authors, thank you very much for the revisions and the additional explanations. However, the fundamental problem that div(c) is in fact not zero (point 3) is still there. Imagine an equilibrium topography with uniform uplift U in 1D. Then Kf*A^m*S^n = U = const and thus d/dx(Kf*A^m*S^n) = d/dx(Kf*Am^m)*S^n + Kf*A^m*d/dxS^n = 0. So your assumption div(c) = d/dx(Kf*A^m) = 0 would imply d/dx S = 0 here, which means that river profiles under uniform uplift would be straight. You introduced an additional argument why div(c) should be zero. This argument is basically that the erosion rate Kf*A^m*S^n can be written as the negative divergence of a sediment flux (-div(Qf)) only if div(c) = 0. It can indeed not be written this way with Qf as a local property, computed from A and S at the considered point. The sediment flux at a given point cannot be computed from A and S at this point for the stream power law because the sediment flux is the integral of the erosion rate over the upstream catchment. So the additional argument cannot enforce div(c) = 0 as we know that it is not zero.

The argument about the propagation of information instead of material does also not imply div(c) = 0. Finally, the problem is that the assumption div(c) = 0 is just at the center of the adjoint method, which is the new aspect introduced in this manuscript. I honestly have no idea how to fix this problem and I am afraid that it cannot be fixed. And I feel that we should not have a wrong assumption right in the middle of a new approach.

Best regards,

Stefan

Re : We thank the reviewer S. Hergarten for giving a very careful attention to our work. We understand why the approach taken by the reviewer sounds like the divergence of the velocity should not be null but it actually misses a very important point: the Stream Power Law (SPL): $E=K_fA^mS^n$ is an equation already obtained by the assumption that the velocity divergence term is null. This is true for both steady-state or transient cases. To prove our point and explain why there was a misunderstanding at the first place we added a new section in the Appendix A: Incompressibility of the Stream Power Law lines 552-571. This section demonstrates why considering the Stream Power Law as a transport equation implies that the velocity at which the topographic information is transported is divergence-free. Questioning this is equivalent to questioning the use of the SPL, which, although legitimate from a very physically fundamental point of view, is out of the scope of our study as we, like many other studies modelling the evolution of the landscape, use the SPL formulation to model fluvial erosion. In addition, divergence-free physically means that the natural decrease of velocity vectors upstream does not result in topographic "accumulation".

In addition, as we stated in our response during the first round of reviews, there are a lot of papers (since the early 70's) which explain why $K_fA^m$ can be interpreted in this equation as a kinematic term, and we think that we can rely on these studies to use the same assumption.

Based on the fact that the SPL + diffusion equation has been very frequently interpreted as a diffusion-advection equation, while this point has never even been questioned elsewhere, we have chosen to add a detailed explanation for this assumption in appendix A at the end of the manuscript. We are however very grateful to S. Hergarten for these remarks, as they have allowed us to dive more deeply into this formulation and its understanding as a diffusion-advection equation.